# Deep quantification of substrate turnover defines protease subsite cooperativity

Rajani Kanth Gudipati [ID][1,2,5], Dimos Gaidatzis[1,3,5], Jan Seebacher [ID][1], Sandra Muehlhaeusser[1], Georg Kempf [ID][1], Simone Cavadini[1], Daniel Hess [ID][1], Charlotte Soneson [ID][1,3] & Helge Großhans [ID][1,4 ✉]

## Abstract

Substrate specificity determines protease functions in physiology and in clinical and biotechnological applications, yet quantitative cleavage information is often unavailable, biased, or limited to a small number of events. Here, we develop qPISA (quantitative Protease specificity Inference from Substrate Analysis) to study Dipeptidyl Peptidase Four (DPP4), a key regulator of blood glucose levels. We use mass spectrometry to quantify >40,000 peptides from a complex, commercially available peptide mixture. By analyzing changes in substrate levels quantitatively instead of focusing on qualitative product identification through a binary classifier, we can reveal cooperative interactions within DPP4's active pocket and derive a sequence motif that predicts activity quantitatively. qPISA distinguishes DPP4 from the related *C. elegans* DPF-3 (a DPP8/9-orthologue), and we relate the differences to the structural features of the two enzymes. We demonstrate that qPISA can direct protein engineering efforts like the stabilization of GLP-1, a key DPP4 substrate used in the treatment of diabetes and obesity. Thus, qPISA offers a versatile approach for profiling protease and especially exopeptidase specificity, facilitating insight into enzyme mechanisms and biotechnological and clinical applications.

**Keywords** DPP-IV Family Proteases; Quantitative Modeling; Peptide Turnover; Protein Engineering; Cryo-EM
**Subject Categories** Methods & Resources; Proteomics

## Introduction

Despite a reputation for being a promiscuous class of enzymes that indiscriminately digest or degrade proteins, many of the ~700 human proteases exhibit substantial substrate specificity. Thus, they can perform diverse functions in the regulation of protein activity, localization, stability and many other molecular and cellular processes important in physiology and disease (Lopez-Otin and Bond, 2008). Accordingly, protease inhibitors and substrate agonists have emerged as important classes of drugs in antiviral therapies as well as for the treatment of non-communicable diseases (Florentin et al, 2022; Leung et al, 2000).

The understanding of the biological functions of proteases, their targeting in clinical settings, and their use in biotechnological applications all require knowledge of their substrates and specificity profiles (Dyer and Weiss, 2022; Vizovisek et al, 2016). According to the nomenclature introduced by (Schechter and Berger, 1967), proteases cleave their substrates by definition between the P1 and the P1' residues (Fig. 1A). Recognition of P1 by a specific active site pocket, generically termed subsite S1, often provides substrate sequence specificity to the enzyme. Substrate residues further to the N-terminus (termed P2, P3, etc.) or to the C-terminus (P1', P2', etc.) of P1 may be recognized by additional subsites (S2, S3 and S1', S2', respectively).

Although this model implies that the interactions between the individual substrate residues and their corresponding subsites are relatively independent of one another, it is evident that constraints in size, geometry or charge may lead to interactions such that certain sequence combinations are favorable, others unfavorable, due to cooperative effects (Ng et al, 2009; Qi et al, 2019) (Fig. 1A). However, quantifying such effects requires large data sets of quantitative cleavage information that have been difficult to obtain. Several mass spectrometry-based techniques such as PICS, TAILS, Subtiligase N-terminomics, COFRADIC, HUNTER, ChaFRADIC (Amiridis and Weeks, 2022; Gevaert et al, 2003; Kleifeld et al, 2011; Kukreja et al, 2015; Schilling et al, 2011; Schilling and Overall, 2008; Venne et al, 2015; Wang et al, 2021; Weeks et al, 2021; Weng et al, 2019) can successfully identify protease-mediated cleavage events in native contexts or in in vitro lysates by enriching and detecting cleavage products. Yet, they tend to be limited to a few hundred sites per experiment (Tsiatsiani and Heck, 2015), and they suffer from an absolute quantification bias of mass spectrometry where the exact sequence of a peptide influences its detectability and quantification (Liigand et al, 2019). These challenges can be partially addressed by more complex experimental designs (Lapek et al, 2019; Plasman et al, 2011), improved enrichment/depletion and labeling procedures (Biniossek et al, 2016; Weng et al, 2019) and greater standardization of input material along with streamlining of processing and improved peptide mapping (auf dem Keller et al, 2010; Uliana et al, 2021) or by reduced complexity of input material through carefully designed

[1]Friedrich Miescher Institute for Biomedical Research, Fabrikstrasse 24, Basel 4056, Switzerland. [2]Center for Advanced Technologies, Adam Mickiewicz University, Uniwersytetu Poznańskiego 10, 61-614 Poznań, Poland. [3]SIB Swiss Institute of Bioinformatics, Basel, Switzerland. [4]Faculty of Natural Sciences, University of Basel, Basel, Switzerland. [5]These authors contributed equally: Rajani Kanth Gudipati, Dimos Gaidatzis. ✉E-mail: helge.grosshans@fmi.ch

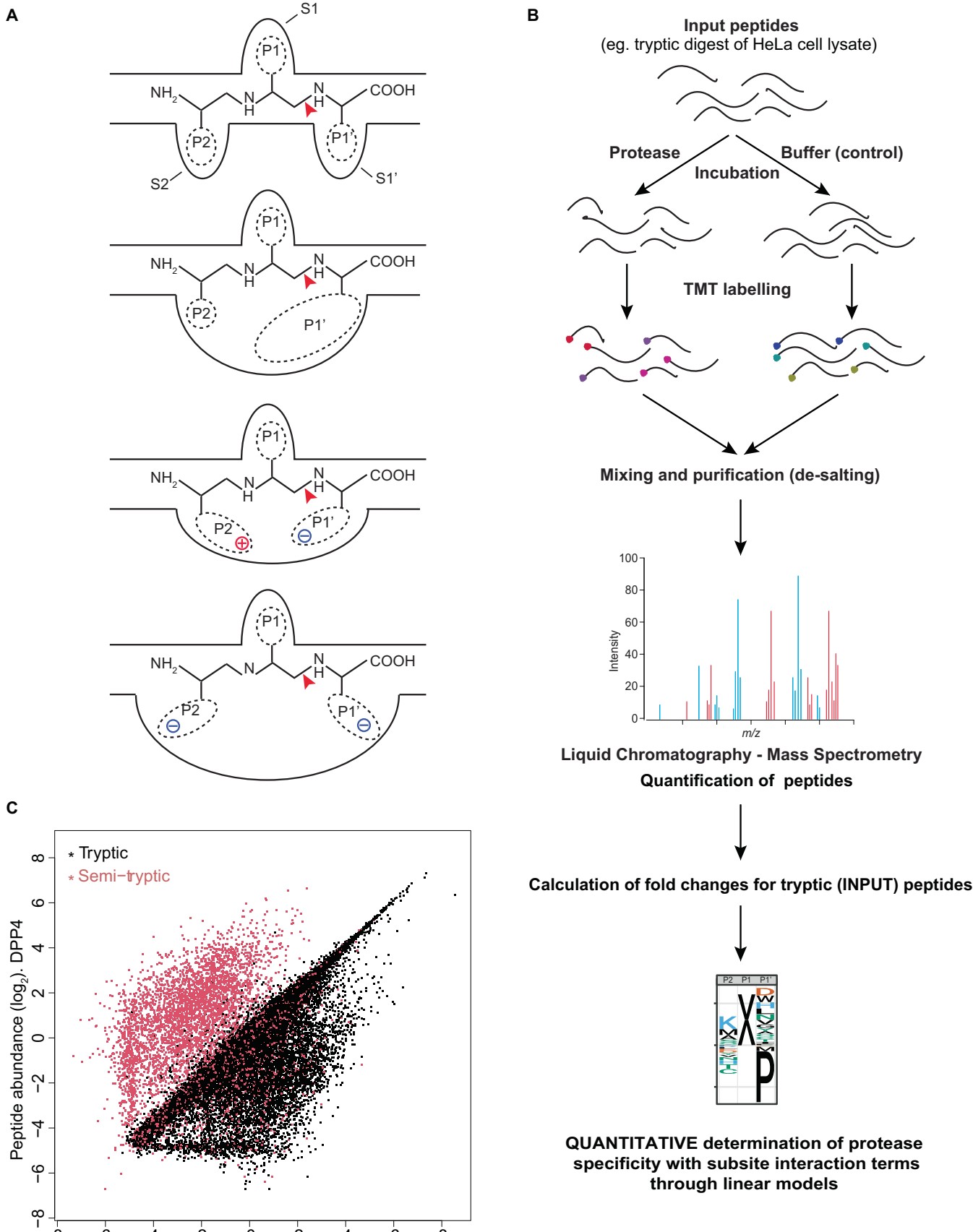

**Figure 1.   Determination of DPP4 cleavage motif.**

(A) Cartoon depicting potential interactions among subsites (S2, S1, S1') in the substrate pocket of a protease and the corresponding substrate positions (P2, P1, P1'). A red arrowhead indicates the scissile bond. (B) Schematic of qPISA. (C) Scatter plot showing median-centered $\log_2$-transformed peptide abundance in DPP4-treated versus buffer control lysates. Each dot represents a unique peptide. Tryptic and semi-tryptic peptides are colored in black and red, respectively.

peptide libraries (O'Donoghue et al, 2012). Yet, the goal of simple and inexpensive generation of quantitative data for a large number of substrates remains to be achieved.

Here, we develop a highly parallelized, mass spectrometry-based approach termed quantitative Protease specificity Inference from Substrate Analysis (qPISA), to quantify the activity of human dipeptidyl peptidase 4 (DPP4) on thousands of peptides. DPP4 is an exopeptidase that regulates blood glucose metabolism through N-terminal processing of the small peptide hormone GLP-1, which initiates GLP-1 degradation and promotes its renal clearance. DPP4 inhibitors and GLP-1 receptor agonists, i.e., derivatives rendered less sensitive to cleavage by DPP4, have thus become two important classes of drugs for the management of type II diabetes and obesity (Deacon, 2019; Palmer et al, 2016). DPP4 clips dipeptides off the N-termini of its substrates (Mentlein, 1999; Walter et al, 1980), with a strong preference for cleavage after proline in the P1 position (Lambeir et al, 2003). Whether and to what extent additional substrate positions contribute to the specificity for DPP4-mediated cleavage has remained unclear. Structural studies have failed to identify specific constraints for these residues (Rasmussen et al, 2003; Thoma et al, 2003), and functional assays have yielded conflicting data, likely due to small numbers of tested substrates and a lack of systematic analysis (Lambeir et al, 2003).

Here, we show that assessing the DPP4 activity on tens of thousands of distinct peptides obtained from a cheap source, HeLa cell tryptic digests, allows us to identify substrate features that promote or impair cleavage, including combinatorial interactions among different substrate positions. The resulting substrate motifs can predict protein half-lives measured in independent studies, and guide approaches to engineer substrates with altered stability. We validate the specificity of the motifs through applying qPISA to another member of the DPP-IV family, *C. elegans* DPF-3, an orthologue of human DPP8/9, which reveals a related but distinct cleavage motif. Through a DPF-3 structure that we solve using cryogenic electron microscopy (cryo-EM), we can rationalize differences in the subsite interactions between the two exopeptidases. Thus, our results broaden our understanding of DPP-IV family proteins by revealing similarities and differences. They provide an unusually comprehensive dissection of subsite cooperativity in proteases and a basis for more targeted approaches to substrate engineering, which we demonstrate in a proof-of-principle experiment for GLP-1. We propose that the procedure that we developed can be readily adapted for the characterization of other proteases.

## Results

### Proteomic analysis reveals abundant peptide-level changes in tryptic HeLa cell lysates treated with recombinant hDPP4

As an important clinical target, hDPP4 has been characterized extensively in research extending over more than three decades. Yet, the peptidase database MEROPS lists only 34 observed

cleavages by hDPP4 (https://www.ebi.ac.uk/merops/cgi-bin/pepsum?id=S09.003;type=P, accessed 9 April, 2024). This dataset supports a clear preference for Pro in P1 (26/34 events), consistent with the fact that the S1 subsite of DPP4 is highly suited to accommodating the unique features of proline, i.e., a pyrrolidine group and a "kink" in the backbone (Thoma et al, 2003). The dataset also implies some benefit of Ala in P1 (4/34 events), again consistent with structural data, but it does not reveal further sequence features that could contribute to specificity. Indeed, no clear picture has emerged from the assessment of individual substrates (Lambeir et al, 2003).

This level of apparent specificity appears surprisingly low, even when considering the possible role of structural features in restricting cleavage to small peptides (Rasmussen et al, 2003). Moreover, many of the known substrates emerged from candidate testing rather than unbiased approaches, leaving it unclear how representative the data is. To generate an unbiased and much larger dataset, potentially capable of revealing combinatorial motifs of sequence specificity and identifying sequence feature both beneficial and detrimental to cleavage, we applied the Proteomic Identification of Cleavage Site Specificity (PICS) (Schilling et al, 2011; Schilling and Overall, 2008) approach (Fig. 1B). Specifically, we utilized a commercially available HeLa cell lysate digested with trypsin as a cheap and diverse source of peptides, that we then incubated with recombinant, commercially available human hDPP4. As pioneered by others (Biniossek et al, 2016; Chen et al, 2017; Tucher and Tholey, 2017), we omitted the enrichment for C-terminal or 'prime-side' cleavage products, and instead directly quantified peptides from treated and untreated control lysates using TMT labeling and mass spectrometry. We could quantify a total of 53,499 unique peptides with a high level of correlation among three replicates (Dataset EV1; Fig. EV1). Based on the known preference of trypsin for cleavage after arginine or lysine, we designated 43,314 peptides (80.96% of detected) as tryptic based on their matching any one of the following three sets of criteria; (i) peptides start downstream of, and end with, Arg/Lys, (ii) peptides start downstream of Arg/Lys and derive from the carboxy terminus of a protein, or (iii) peptides end with Arg/Lys and derive from the amino terminus of a protein. These are the input peptides. The remaining 10,185 (19.04%) semi-tryptic peptides represent potential products of the digestion with DPP4. Supporting this assignment, a comparison of peptide intensities before and after digestion revealed that most of the tryptic peptide intensities were unaffected or reduced after protease incubation, while semi-tryptic peptides mainly increased in signal intensity (Fig. 1C).

### Substrate-focused analysis identifies beneficial residues in the P1 position of DPP4 substrates

PICS and related approaches are aimed at enriching and detecting cleaved peptides upon exposure of peptide mixtures to proteases of interest (auf dem Keller et al, 2010; Biniossek et al, 2016; Schilling et al, 2011; Schilling and Overall, 2008). The focus on product

detection is beneficial because product detection can directly ascertain cleavage. However, it is poorly suited to quantitative analysis because of both biases in the detectability of individual peptides from complex mixtures (Liigand et al, 2019; Mallick et al, 2007) and the lack of baseline levels to which to relate product peptide amounts. Accordingly, data are analyzed through a binary classifier that distinguishes between products and non-products while ignoring quantitative differences in cleavage efficiencies.

A direct quantification of input peptides could potentially circumvent these limitations because these peptides are present in both protease-treated and control lysates, allowing for a straight-forward quantification of differential abundance. This analysis would also compensate for certain compositional biases in the input material (such as the HeLa tryptic digests used by us), which would be canceled out (see "Discussion"). Hence, by focusing on the substrates, we saw an opportunity to build a continuous predictor that not only distinguishes substrates from non-substrates but, uniquely, also captures the extent of proteolysis as a measure of activity. To this end, we devised a linear model to predict the tryptic substrate peptide intensity changes in the experiment from their amino acid sequences. Linear models (described in more detail in "Methods") use a set of independent variables as input to calculate an output such that the correlation between the output of the model and the observed data is maximized. This approach does not require the selection of potential substrates through a statistical test to distinguish changing peptides from non-changing peptides in a binary fashion. Instead, the model directly predicts the $\log_2$ fold change between the buffer control samples and the enzyme-treated samples. Its performance can be quantified by the $R^2$, which is the amount of variability in the data that it can explain. In the following, to allow for a fairer comparison among models of different complexities, we use the adjusted $R^2$, which accounts for different numbers of parameters in the models.

To determine the most important positions within the peptides, we evaluated a series of simple models that each used only one out of the six positions from P2 to P4' as a single predictor. This revealed that the identity of the amino acid at P1 (the second residue from the N-terminus) is a key specificity feature, accounting for a substantial 60.7% of the total variance of the peptide changes observed in the experiment (Fig. 2A). Contrasting with the large contribution from P1, the next two most highly scoring positions, P2 and P1', accounted for only 1.4% and 1.2% of the total variance, respectively. Indeed, the S2 and S1' subsites are relatively open and can accommodate also bulky side chains, explaining the moderate contribution to specificity (Rasmussen et al, 2003; Thoma et al, 2003).

The other three positions (P2'–P4') were essentially negligible at ≤0.5% each, consistent with structural observations of DPP4 in complex with a decapeptide, which revealed that only residues from P2 to P2' form specific interactions with DPP4 side chains (Aertgeerts et al, 2004). Even when we considered the first three (P2, P1, and P1') or all six positions simultaneously, in a linear fashion, the predictive power was only slightly increased to 63.1% and 63.5% of total variance, respectively.

We used the model that combined all six positions in a linear fashion to generate a sequence logo (Fig. 2B). As expected, this revealed proline in P1 as the most beneficial amino acid. Surprisingly, however, alanine appeared similarly beneficial. This

was unexpected since proline had been reported to outperform any other residue in P1 regarding both substrate fit to the S1 subsite (Thoma et al, 2003) and cleavage (Lambeir et al, 2003). It was also unexpected given the much lower frequency of alanine than proline at the P1 site of substrates reported in the MEROPS database. However, the observation is entirely consistent with the fact that key physiological substrates of DPP4, namely GIP-1 and GLP-1$_{7-37}$, contain alanine in P1. Alanine contains only a methyl group as a side chain, which provides two features potentially beneficial for cleavage by DPP4: a small size that is well tolerated in the S1 subsite and a reduction of allowed backbone torsion angles, which facilitates the formation of a kinked backbone conformation (Aertgeerts et al, 2004; Rasmussen et al, 2003; Thoma et al, 2003).

Consistent with previous observations (Martin et al, 1993), Ser and Thr also appeared beneficial for cleavage in P1, but to a much lower extent than Ala or Pro. This may be explained by their reaching the size tolerance threshold of the S1 subsite (Rasmussen et al, 2003). Finally, some residues in P2 and P1' appeared moderately detrimental to cleavage, notably proline in P1'.

## hDPP4 target site motifs reveal context dependency of residues in particular substrate positions

Although the outstanding importance of P1 revealed by our analysis agrees well with the published literature, it left a large fraction of the variance—and thus the features that determine substrate quality—unaccounted. We suspected that certain combinations of residues could be particularly beneficial for, or detrimental to, cleavage. Specifically, a crystal structure of DPP4 in complex with a substrate revealed that the side chains of P2 and P1' point toward each other (Thoma et al, 2003), suggesting that certain rotameric combinations might cause clashing, attraction, or repulsion, and thereby affect substrate accommodation in a hydrolysis-competent conformation (Fig. 1A).

The large number of substrates for which we had quantitative data allowed us to systematically test the occurrence of this and other interactions. Thus, we created a set of additional models that contained pairwise interaction terms in addition to a linear combination of the first three positions. Surprisingly, inclusion of P2:P1' interactions increased the predictive power of the model by merely ~0.4 percentage points relative to the baseline model with the linear combination of the first three positions. By contrast, incorporating interactions between either positions P2 and P1 (P2:P1) or P1 and P1' (P1:P1') increased the predictive power of the model by 3.4 and 6.0 percentage points, respectively. This was equally unexpected given that the structural data had shown that the P1 side chain is shielded inside the S1 subsite from those of its two neighbors P2 and P1', and that it additionally points in an opposite direction from them (Thoma et al, 2003). Nonetheless, a model including the P2:P1 and the P1:P1' interaction terms simultaneously increased the predictive power of the model by 9.6 percentage points relative to the scenario that considered all positions independently, and by 12 percentage points relative to the "P1 only" baseline scenario. Thus, the final model achieved a predictive power of 72.7%.

The increase in predictive power supports the existence of complex interactions among the three substrate positions that together define substrate specificity. To visualize the rich results of the final model for DPP4, we created multi-panel sequence logos,

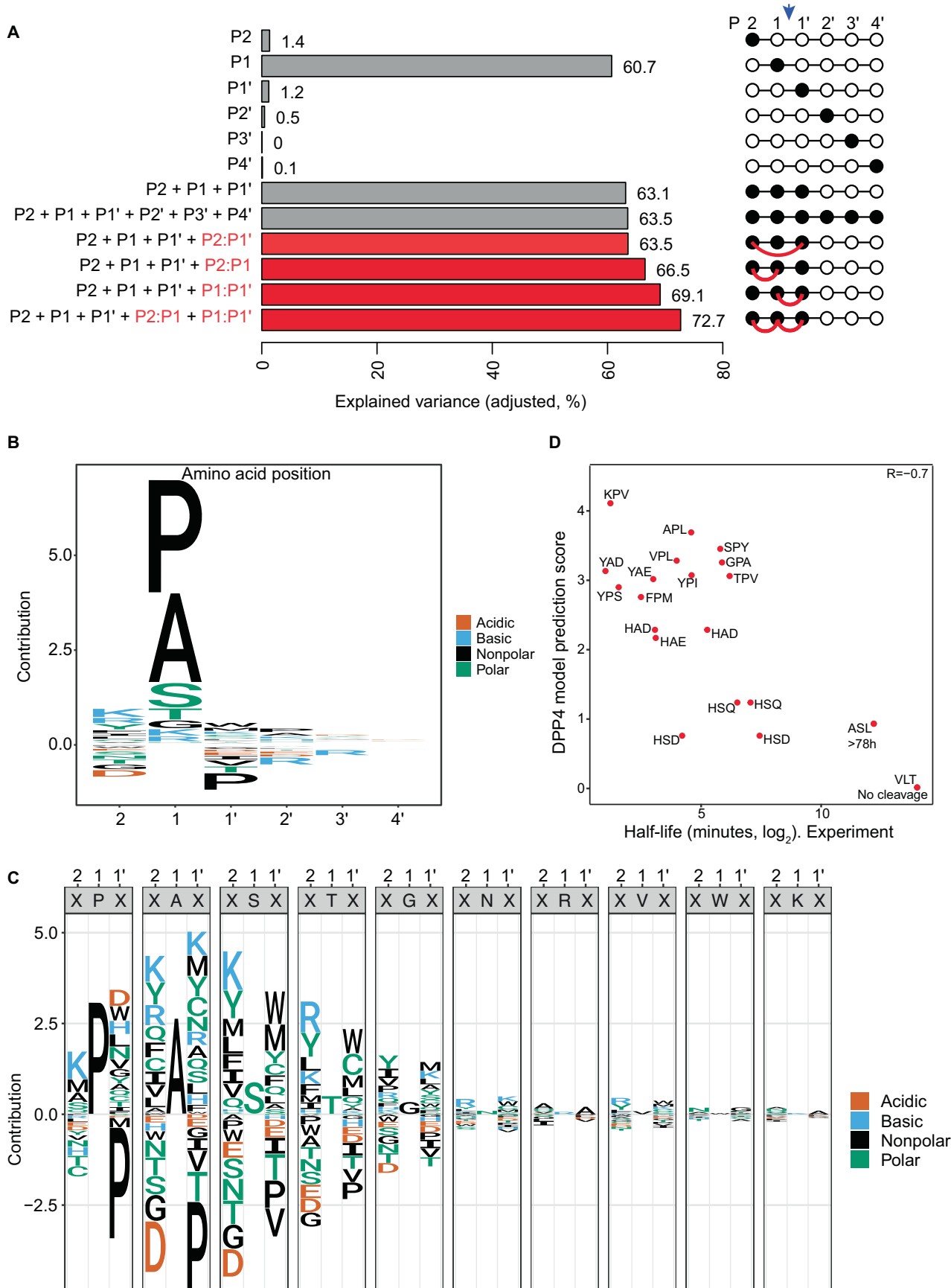

**Figure 2.   Models depicting DPP4 cleavage specificity.**

(A) Bar plots showing the predictive power (variance explained for tryptic peptides) of a total of twelve linear models, each incorporating a specific set of predictors, indicated on the left and schematically displayed on the right. The red connecting line indicates interaction terms between the respective amino acids. A downward arrow indicates the cleavage site. The performance of each model is quantified via the adjusted $R^2$ to account for differences in the numbers of parameters among the models. (B) Sequence logo generated by the model using the identity of the N-terminus (first 6 amino acids) of all quantified tryptic peptides. The height of each amino acid letter in the sequence logo represents the efficiency of the contribution towards cleavage either positively (above zero) or negatively (below zero). (C) Sequence logo representing the cleavage specificity of DPP4 inferred by a linear model (A, bottom). For each amino acid at position P1, a sequence logo triplet depicts the cleavage specificity considering the contributions of P1 as well as the two neighboring (P2 and P1') amino acids. The height of each amino acid letter in the sequence logo represents the contribution to cleavage efficiency. Positive numbers indicate an increase in cleavage efficiency, while negative numbers denote a reduction in cleavage efficiency. The total efficiency of a triplet is given by the sum of the contributions of the amino acids occupying three (P2, P1, and P1') positions. (D) Scatter plot of experimentally determined peptide half-lives upon incubation with DPP4 (from (Keane et al, 2011)) vs. qPISA model substrate score. Each dot is a unique peptide whose first three amino acids are indicated.

depicting the contributions of positions P2 and P1' given a certain amino acid in P1 (Fig. 2C; Dataset EV2). This allowed us to visualize the full model, including the numerous interaction terms ("Methods"). The sequence logos depict positive as well as negative contributions of specific amino acids, relative to "average" cleavage, with the value of P1 containing the positional identity term of P1 and the values of P2 and P1' their respective positional identity terms plus their respective interaction terms with P1. Thus, the total predicted cleavage efficiency for a specific amino acid triplet is given by the sum of all three individual contributions, which may be positive or negative.

Consistent with the analysis above, Pro and Ala in P1 appeared by far the most important cleavage-promoting feature, while Ser and Thr were also beneficial, albeit less so. At the other two positions, Pro in P1' emerged as the most consistent feature, being highly detrimental in the combination with all relevant P1 residues and the largest negative predictor of cleavage from our model. Indeed, although the simpler model lacking interactions pointed to a modest negative contribution of Pro in P1' (Fig. 2B), the interaction model reveals that Pro in P1' can effectively neutralize even the positive contribution of Pro and Ala at P1. This is explained by the fact that proline at P1' would induce another kink in the peptide backbone which is incompatible with the substrate binding mode. It is also consistent with previous work that failed to observe cleavage of certain P1'(Pro)-containing peptides by hDPP4 (Puschel et al, 1982).

Thus, proline in P1' exhibited a quantitative context-dependent effect in that the size of its negative effect depended on the size of the positive effect of the residue in P1. Other residues in either P2 or P1' revealed even qualitative P1-dependent effects. For instance, Thr and Val in P1' were mildly beneficial for cleavage when P1 was occupied by Pro but clearly detrimental with Ala, Ser, or Thr in P1. Such context-dependence agrees with a lack of substantial predictive power from individual positions other than P1, or linear combinations of P2, P1, and P1', described above.

## The DPP4 model predicts substrate turnover kinetics

Given that the subsite interaction data revealed through our analysis could not be predicted from available structural data (Thoma et al, 2003), we sought to validate our findings and test whether the model could predict cleavage activity on an unrelated dataset. To this end, we used a published dataset providing the half-lives of 21 synthetic peptides (representing physiological and pharmacological substrates of DPP4) upon incubation with DPP4 in vitro (Keane et al, 2011). Strikingly, although our model was

derived from analysis of peptide turnover at a single time point, and thus did not contain any kinetic information, we observed a very good anticorrelation (Pearson $R = -0.7$) between our model prediction score and the previously reported half-lives (Fig. 2D). Thus, peptides predicted to be good substrates according to a high model score had short half-lives, whereas low-scoring peptides had longer half-lives. Notably, this analysis also confirmed that in the context of extended peptides, alanine in P1 can indeed be as beneficial for cleavage as proline, exemplified by GRF-amide, which starts with the tripeptide YAD, as one of the most rapidly processed substrates of DPP4 in vitro. We conclude that qPISA can predict cleavage efficiencies quantitatively.

## Substrate versus product: a comparison

Our mass spectrometric analysis identified not only >43,000 tryptic (input) peptides but also 3600 semi-tryptic peptides that accumulated >twofold in DPP4-treated vs. control lysates, identifying them as potential cleavage products. We wanted to test whether we could also use these peptides to obtain quantitative insight into DPP4 specificity using a similar modeling approach. To this end, we used the tryptic peptides that did not change in intensity upon incubation as the background for the model ("Methods"). These peptides reflect the skew in amino acid composition in the HeLa proteome, exhibit K/R depletion, and account for the detection bias of the mass spectrometer (see "Discussion").

Linear modeling generated results that were qualitatively similar, yet quantitatively markedly weaker than the results obtained from modeling substrate peptides (Fig. 3A). Thus, P1 stood out in explaining 44.1% of total variance on its own (compared to 49.6% when using substrates), and the best performing model, using P2, P1, P1' along with P2:P1 and P1:P1' interaction, explained 54.7% of the total variance (compared to 72.7% when using substrates). Specifically, when compared to qPISA, this analysis confirmed similar benefits of Ala and Pro in P1 (Fig. 3B). However, clear differences emerged in the interaction term parameters (Fig. 3C), and the product-based model predicted the previously measured peptide half-lives less well (Fig. 3D), especially for peptides containing P1 Pro. We conclude that the linear modeling approach can also be applied to product peptides to obtain quantitative information on cleavage efficiencies, but that modeling using substrates is both more straightforward (because no separate steps need to be taken to identify a suitable background; see also "Discussion") and more powerful.

qPISA is particularly well suited for exopeptidases, where the cleavage position on a substrate is defined (e.g., two amino acids

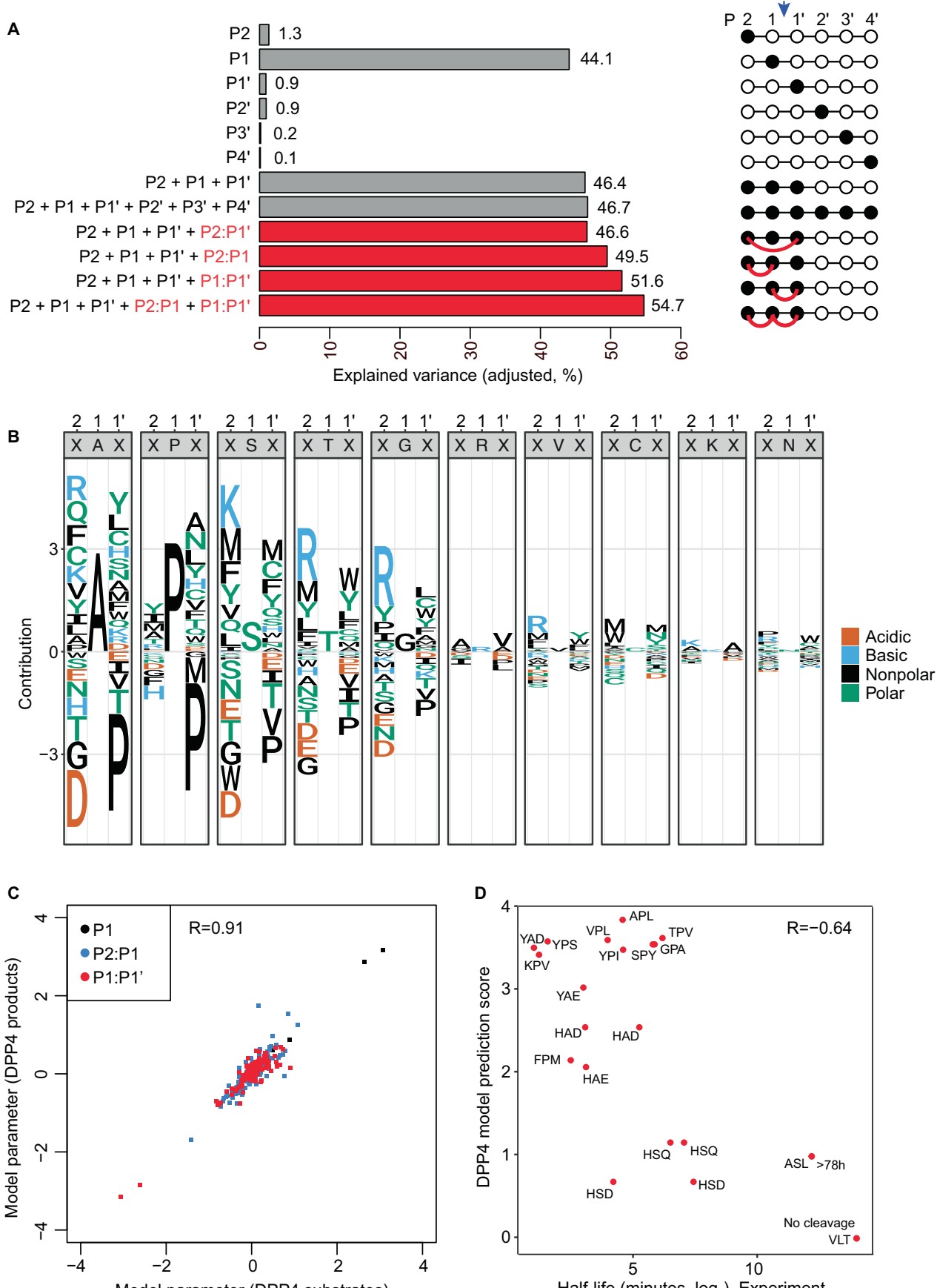

**Figure 3. Models built on product peptides depicting DPP4 cleavage specificity.**

(A) Bar plots showing the predictive power (variance explained for semi-tryptic (product) peptides) of a total of twelve linear models, each incorporating a specific set of predictors, indicated on the left and schematically displayed on the right. The red connecting line indicates interaction terms between the respective amino acids. A downward arrow indicates the cleavage site. The performance of each model is quantified via the adjusted $R^2$ to account for differences in the number of parameters among the models. (B) Sequence logo representing the cleavage specificity of DPP4 inferred by a linear model considering semi-tryptic (product) peptides (A, bottom). For each amino acid at position P1, a sequence logo triplet depicts the cleavage specificity considering the contributions of P1 as well as the two neighboring (P2 and P1') amino acids. The height of each amino acid letter in the sequence logo represents the contribution to cleavage efficiency. Positive numbers indicate an increase in cleavage efficiency while negative numbers denote a reduction in cleavage efficiency. The total efficiency of a triplet is given by the sum of the contributions of the amino acids occupying three (P2, P1, and P1') positions. (C) Comparison of models generated using semi-tryptic peptides (DPP4 products; $y$ axis) or tryptic peptides (input; $x$ axis). Each dot represents one model parameter; Pearson correlation is indicated. (D) Scatter plot of experimentally determined peptide half-lives upon incubation with DPP4 (from (Keane et al, 2011)) vs. qPISA model score obtained by analyzing the semi-tryptic (product) peptides. Each dot is a unique peptide whose first three amino acids are indicated.

away from the N-terminus, in the case of DPP4). For endopeptidases, this information is not immediately available. However, it can be obtained from product peptides detected in the same experiment so that the cleavage site can be mapped onto the matching tryptic substrate peptide. As this approach would require a sufficiently large number of matched pairs of product and substrate peptides, we examined the extent of overlap in our assay. Notably, for the 3600 tryptic peptides that we had identified as products, we found 2348 matching substrates (65.2%), a number that is likely to increase with future improvements in the detection efficiency of mass spectrometers. Hence, we can obtain quantitative cleavage information from substrate peptide analysis while sacrificing only 34.8% of the detected cleavage events, indicating that qPISA can be beneficial also for the quantitative characterization of endopeptidases.

## Comprehensive qPISA data provide a basis for protein engineering

Human DPP4 is a key regulator of blood glucose level, which it controls chiefly through the processing of the incretin Glucagon-Like Peptide (GLP-1$_{(7-37)}$). The processed GLP-1$_{(9-37)}$ peptide exhibits reduced affinity to its receptor and becomes a target of rapid renal secretion and subsequent degradation, reducing its half-life to approximately two minutes (Mentlein et al, 1993; Muller et al, 2019). Accordingly, DPP4 inhibitors and GLP-1$_{(7-37)}$ receptor agonists (GLP-1RA) have become important drugs for the treatment of type 2 diabetes (Palmer et al, 2016). To achieve clinical utility, GLP-1RAs such as semaglutide required stabilization against DPP4-mediated cleavage, and modifications that include the replacement of Ala in the P1 position with Gly or 2-aminoisobutyric acid helped to increase the half-life to a maximum of about 165 h (Lovshin, 2017). Since further GLP-1RA stabilization would simplify medication schedules that require subcutaneous injections, there is a continued interest in further increasing its half-life without compromising its agonist function. However, the protein engineering space is limited by the fact that the amino terminus of GLP-1$_{(7-37)}$ is deeply embedded in the GLP-1 receptor transmembrane core and required for full receptor activation (Zhang et al, 2017). Hence, the ability to modulate "cleavability" at additional sites may help to meet the competing requirements of stabilization and maintenance of biological activity.

To provide proof-of-principle that our approach can be used to expand the substrate engineering space on a peptide of interest, we assayed the activity of hDPP4 on different 12 residue-long synthetic peptides representing the N-terminus of GLP-1$_{(7-37)}$ and variants thereof. As GLP-1$_{(7-37)}$ starts with HAE, we investigated the effect of altering P2 (from His to Asp) and/or P1' (from Glu to Pro), two of the most inhibitory amino acids toward cleavage for peptides containing Ala in P1 according to our model (Fig. 2C, second panel). Product formation from the canonical substrate (HAE) in an in vitro assay was essentially complete within ≤30 min (Fig. 4A). By contrast, replacement of His at P2 with Asp (DAE) yielded greatly decreased product levels at all time points and continued product accumulation over 120 min, consistent with the model prediction of reduced DPP4 activity on this substrate (Fig. 4A). No measurable product generation was observed in buffer-only control reactions (Fig. 4B). In addition, and again consistent with the model prediction, replacement of glutamate in P1' with proline (either alone, in the HAP peptide, or together with aspartate in P2, DAP) completely abrogated product formation (Figs. 4A,B and EV2; Dataset EV6).

The introduction of proline in P1' may destabilize the GLP-1 helix fold, which is important for receptor binding, thus negatively affecting activity. However, molecular modeling, using the GLP-1 structure bound to its receptor described in (Zhang et al, 2021), suggested that aspartate in P2 will likely be well tolerated (Fig. 4C). Hence, these data indicate that qPISA can be used to rationally design protease substrates with desired properties.

## C. elegans DPF-3 differs in specificity from hDPP4

To examine the substrate specificity of a second DPP-IV family exopeptidase, we selected C. elegans DPF-3, a DPP8/9-orthologue that promotes male fertility through N-terminal processing of two large RNA-binding proteins, WAGO-1 and WAGO-3 (Gudipati et al, 2021). We repeated qPISA using recombinant DPF-3 produced by baculovirus-mediated expression in *Trichoplusia ni* High-Five insect cells. As for DPP4, most non-tryptic peptides showed a substantial increase in signal intensity after incubation with DPF-3 relative to a buffer control, whereas most tryptic peptides had unchanged or decreased levels (Fig. 5A; Dataset EV1).

We applied the same linear model analysis to DPF-3, thereby revealing a similar set of substrate specificity features, i.e., a large predictive power of P1 (49.6% of total variance), little contribution of any of the other positions individually or jointly, but a substantial increase (to 67.1%) when introducing the two interaction terms P2:P1 and P1:P1' (Fig. 5B; Dataset EV3).

As in our analysis of hDPP4, we used the model that combined all six positions in a linear fashion to generate a sequence logo to indicate the amino acid preference of DPF-3 (Fig. 5C). As expected,

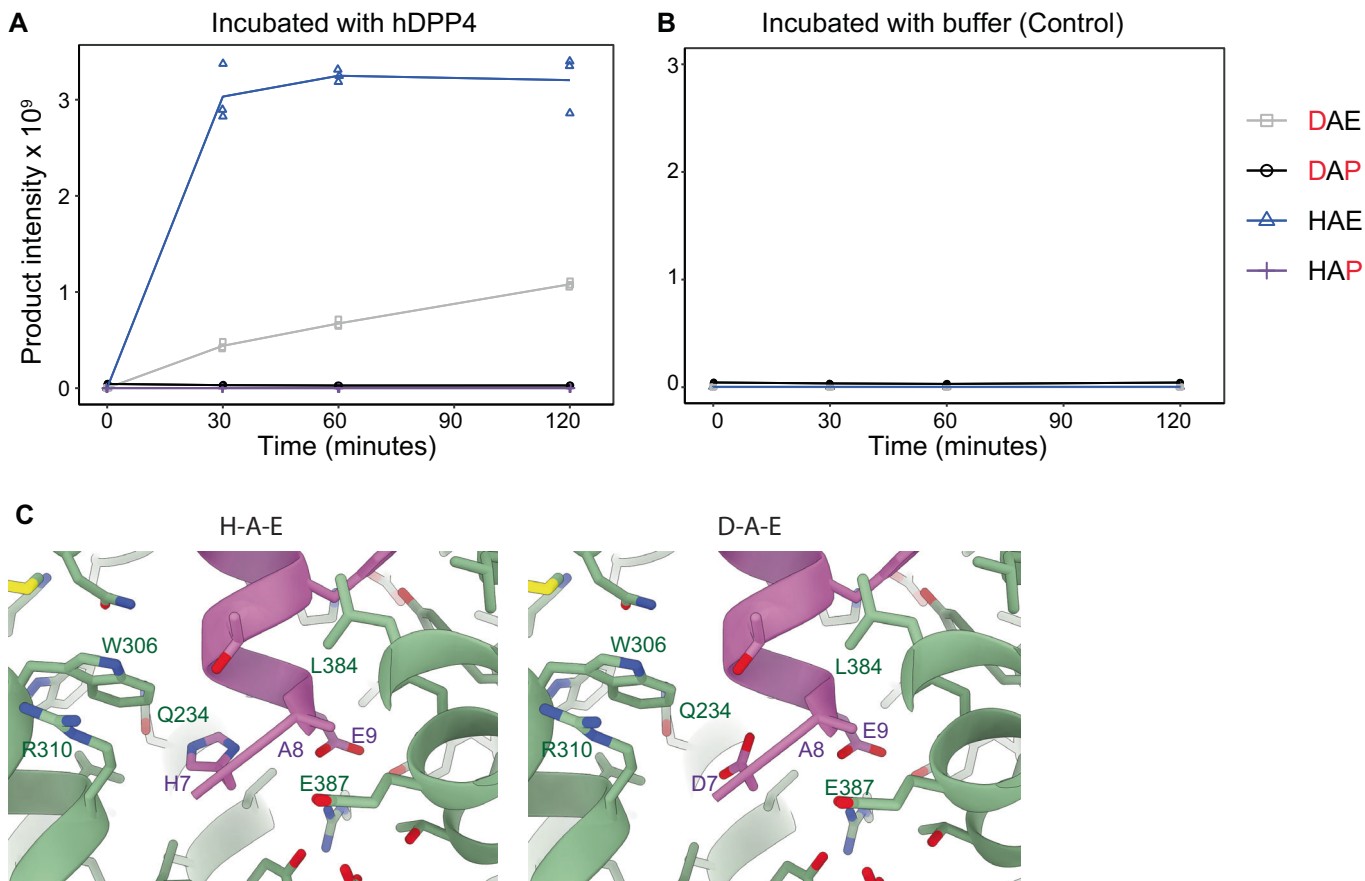

**Figure 4. The model predicts protease activity.**

(A, B) Product peptide intensity of a GLP-1$_{(7-36)}$-derived N-terminal peptide (HAEGTFTSDVSR) and its variants plotted over reaction time upon digestion with hDPP4 (A) or incubation with buffer only (B). The substrate peptides are identical in positions 4–12 and differ in position 1 and/or 3 as indicated in the respective product labels. The expected products are EGTFTSDVSR or PGTFTSDVSR. To distinguish the product generated from a given substrate, the four peptides were pooled into two groups. One group contained HAE, DAP and the other group had DAE and HAP. Data points of three technical replicates are shown; a line indicates mean values. (C) Left panel, N-terminus of GLP-1$_{(7-36)}$ (purple) bound to GLP-1 receptor (green, PDB ID6X18, (Zhang et al, 2020). Right panel: model for GLP-1$_{(7-36)}$ peptide with histidine 7 (position P2) replaced by aspartate (structure from left panel used as template, see "Methods").

this revealed a strong benefit of proline in P1. Contrasting with hDPP4, however, alanine had a much smaller benefit in this position.

We visualized the full model, including the numerous interaction terms, as described above for hDPP4. In agreement with the results for DPP4, we found that Pro, Ala, Thr, and Ser in P1 were all favorable for substrate cleavage (Fig. 5D). However, when we applied the model to predict the half-lives of hDPP4 substrates, it performed substantially less well (Pearson $R = -0.41$ vs. $R = -0.7$ for the hDPP4 model; Fig. EV3A). This finding suggested substantial differences between the substrate specificities of DPF-3 and DPP4. Confirming the robustness of the results, we repeated DPF-3 qPISA (Dataset EV4) and obtained highly similar model parameters (Pearson $R = 0.96$, Fig. 6A; Dataset EV5). By contrast, the correlation was much lower ($R = 0.76$) with the parameters of the DPP4 model (Fig. 6A).

To visualize the differences in the specificities of DPF-3 and DPP4, we subtracted the model parameters and generated a new sequence logo depicting the differences between the two enzymes. This analysis confirmed the greater preference of DPP4 vs. DPF-3

for Ala, Ser and Gly at P1 (Fig. EV3B) and revealed extensive differences for various amino acid combinations. Thus, Asp or Glu in P2 were broadly detrimental for DPF-3-mediated cleavage, but much more context-dependent for DPP4 (Figs. 2C, 5D, and EV3B). For instance, for DPP4, both amino acids appeared largely neutral when Pro occupied P1, yet Asp but not Glu was detrimental when Ala occupied P1. In P1', both enzymes prefer aromatic or hydrophobic residues, but the effect is generally stronger for DPF-3 than DPP4, and for both enzymes less pronounced when P1 is occupied by Pro than any other amino acid.

Finally, we asked if the two models could detect differences in the specificity of DPF-3 and DPP4 at the single peptide level. To this end, we predicted the cleavage efficiency for all the detected tryptic peptides using both models separately and compared the two output scores (Fig. 6B). This showed that one group of peptides was predicted to undergo efficient cleavage according to both models, while specific subsets of peptides were predicted to undergo preferential cleavage by either DPF-3 or DPP4. Overlaying those predicted subsets of peptides with the experimentally determined abundance changes for each detected tryptic peptide showed a high

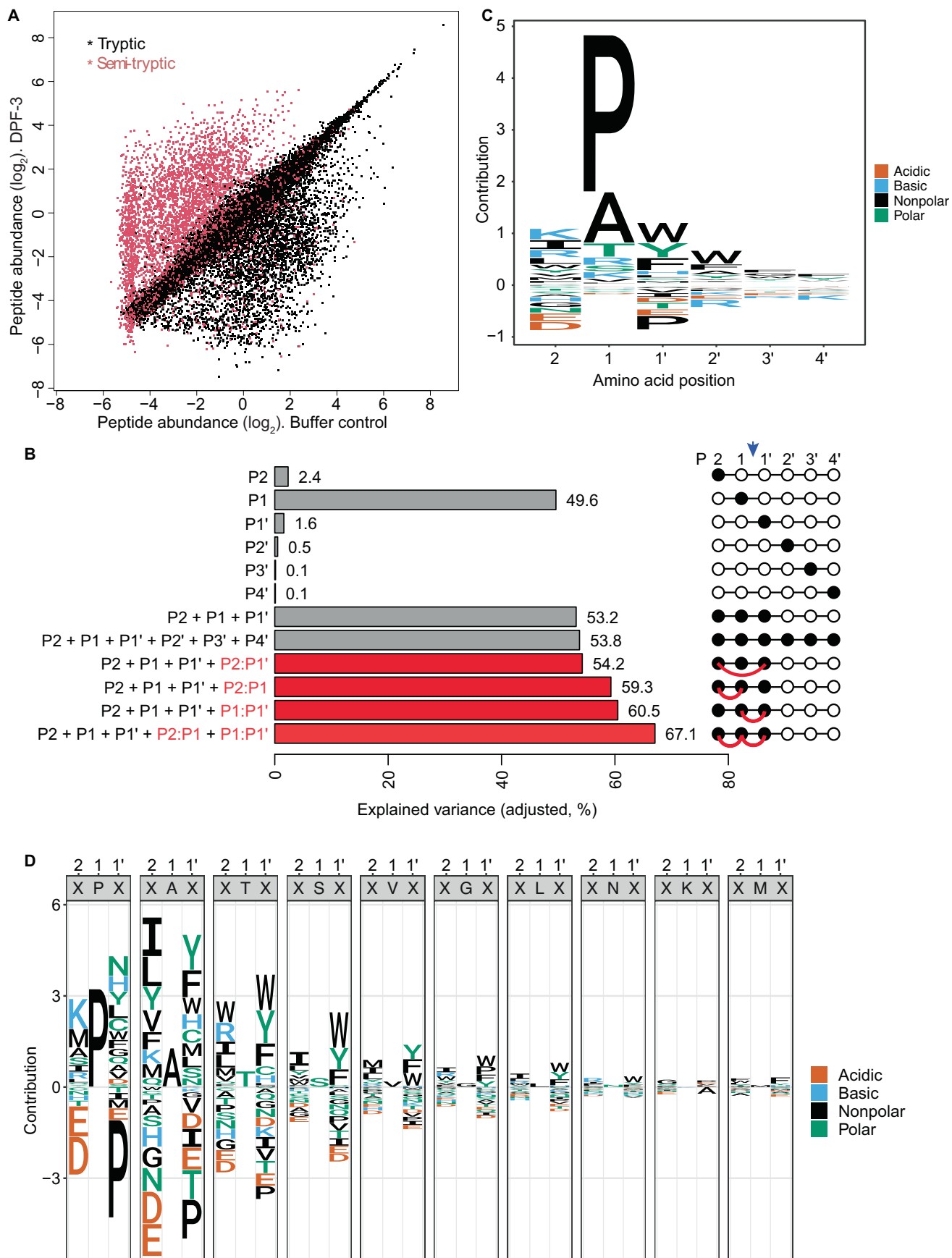

◀ **Figure 5.  Determination of the DPF-3 cleavage motif.**

(A) Scatter plot showing median-centered log$_2$-transformed peptide abundance in DPF-3-treated versus buffer control lysates. Each dot represents a unique peptide. Tryptic and semi-tryptic peptides are colored in black and red, respectively. (B) Bar plots showing the predictive power (variance explained) of a total of twelve linear models, each incorporating a specific set of predictors, indicated on the left and schematically displayed on the right. The red connecting line indicates interaction terms between the respective amino acids. A downward arrow indicates the cleavage site. The performance of each model is quantified via the adjusted $R^2$ to account for differences in the number of parameters among the models. (C) Sequence logo generated by the model using the identity of the N-terminus (first 6 amino acids) of all the DPF-3 substrates. The height of each amino acid letter in the sequence logo represents the efficiency of the contribution towards cleavage either positively (above zero) or negatively (below zero). (D) Sequence logo representing the cleavage specificity of DPF-3 inferred by a linear model (B, bottom). For each amino acid at position P1, a sequence logo triplet depicts the cleavage specificity considering the contributions of P1 as well as the two neighboring (P2 and P1′) amino acids. The height of each amino acid letter in the sequence logo represents the contribution to cleavage efficiency. Positive numbers indicate an increase in cleavage efficiency while negative numbers denote a reduction in cleavage efficiency. For P2 and P1′, values comprise both the positional identity and the respective interaction terms with P1; the model intercept term is integrated into the P1 value. Thus, the total efficiency of a triplet is given by the sum of the contributions of the amino acids occupying the three (P2, P1, and P1′) positions.

agreement, indicating that the two models can indeed distinguish the specificity of the two enzymes at the single peptide level.

## A cryo-EM structure of DPF-3 reveals distinct S2 and S1′ subsite features

To relate the observed differences in substrate specificities of DPF-3 and DPP4 to structure, we determined the structure of DPF-3 to an effective resolution of 2.6 Å by single particle cryogenic Electron Microscopy (cryo-EM). The map was interpretable for residues 37–123, 259–431, and 470–916. The resolution varied between 2 Å and 6 Å (Figs. 7A, EV4 and 5; Dataset EV7). The overall structure follows the conserved dipeptidyl peptidase (DPP) fold that is characterized by two domains, a C-terminal hydrolase with α/β fold, and an N-terminal β-propeller domain encompassing eight blades (Fig. 7B) as observed in other DPPIVs (Ross et al, 2018). DPF-3 superposes with the apo structures of its orthologues DPP8 and DPP9 with r.m.s.d. (root mean square deviation) values of 3.4 and 3.0 Å (dimer) as well as 3.6 and 2.9 Å (monomer), suggesting strong structural conservation.

A comparison of the DPF-3 apo structure with DPP4 reveals an S1 subsite that is nearly identical for both enzymes, with all amino acids being conserved (Fig. 7C). Moreover, no outstanding differences regarding residue positions or side chain conformations were evident in the structure to explain altered specificities between DPF-3 and DPP4. Instead, the divergent enzymatic footprints appear best explained by differences in the S2 and S1′ subsites. Unlike DPP4, DPF-3 contains an acidic residue (E309) but lacks a bulged loop inserting a hydrophobic/aromatic residue (F357 in DPP4) into the S2 subsite, which alters its geometry and chemical environment substantially. E309, which structurally aligns with a serine in DPP4 (S209), is adjacent to the double glutamate motif that is required to position the terminal NH3$^+$ group (Thoma et al, 2003).

This difference between the two proteins is consistent with the stronger negative contribution of acidic residues, i.e., Asp and Glu, at P2 in DPF-3 relative to DPP4. A decreased tolerance for certain amino acids in S2 may also require a greater contribution from the other subsites for efficient cleavage, thus offering a potential explanation of DPF-3's clear preference for P1(Pro) with its optimal S1 subsite binding and the constrained substrate backbone.

Another significant difference from DPP4 is the presence of an isoleucine (I705) in the S1′ subsite of DPF-3 that is oriented towards the P1′ substrate position. In DPP4, the corresponding side chain is a serine (S552), which is retracted from the substrate along with a segment of the loop. This difference is consistent with DPF-3's enhanced preference for several hydrophobic and all aromatic

amino acids at P1′, because the isoleucine may facilitate hydrophobic packing of these side chains to contribute to substrate binding or positioning.

## Discussion

Detailed knowledge of substrate specificity and its underlying mechanisms not only reveals physiological and pathological functions of proteases, but also facilitates their clinical and biotechnological application, for instance by instructing protein engineering approaches to optimize protease substrates. The qPISA approach that we developed here helps to generate such knowledge, allowing the identification of combinatorial substrate motifs, including positive and negative contributions, for exopeptidases.

### Experimental and analytical advances to identify protease substrate motifs

Our analysis combines a cost-effective and readily available source of large numbers of diverse peptides (~$4 \times 10^4$ in our analysis) with a quantitative read-out of activity by mass spectrometry. The approach does not require any enrichment/depletion procedures and associated investments in input material, time, money and specialized expertise and/or materials. By focusing on substrates, a combination of large-scale quantitative data with mathematical modeling permits us to go beyond identification of individually important substrate positions and beneficial residues at that position, to uncovering cooperative effects arising from combinations of residues. Moreover, in contrast to traditional binary classifiers that distinguish qualitatively between substrate and non-substrate, qPISA provides quantitative insight into cleavage efficiencies across the input peptide sequence space.

A major benefit of qPISA's focus on substrates is that the tryptic peptides that do not change in intensity serve as an internal background in the modeling process, which can account for several experimental biases: First, the HeLa proteome has a biased amino acid composition, e.g., L has a prevalence of 9.9%, W of 1.2%. This issue is further exacerbated when considering neighboring pairs of amino acids (as quantified by interaction terms), e.g., LL has a prevalence of 1.1%, WW only 0.02%. Second, the peptides in the HeLa tryptic digest are strongly depleted for K and R as a result of trypsin's specificity. Finally, peptides have varying detection rates in the mass spectrometer given their amino acid composition. Without a proper background, these biases would immediately skew quantitative assessment of enzyme specificity, in particular for the interaction terms. An

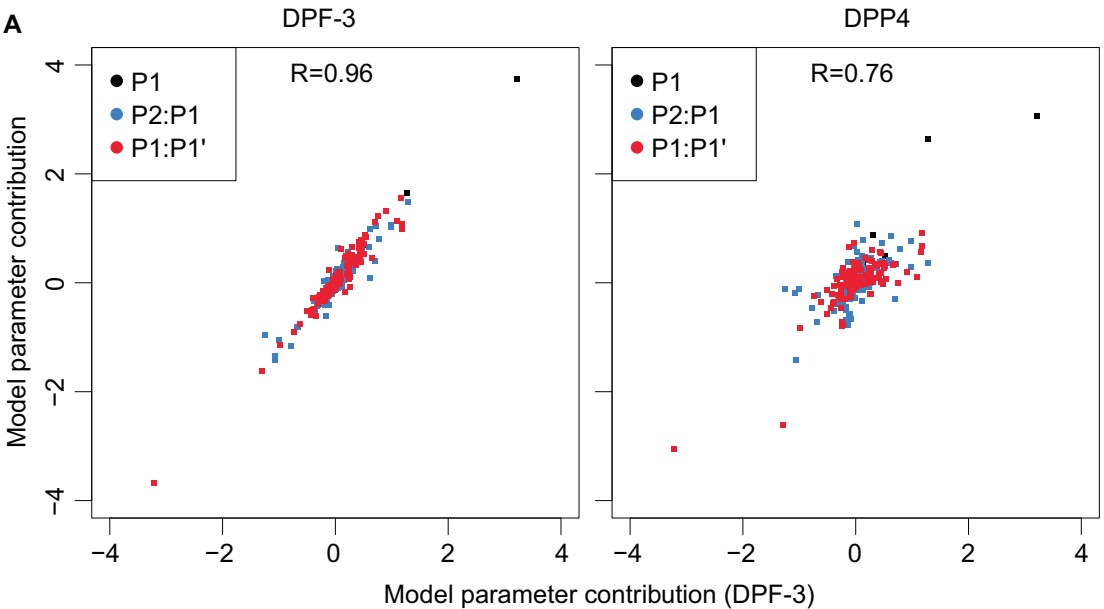

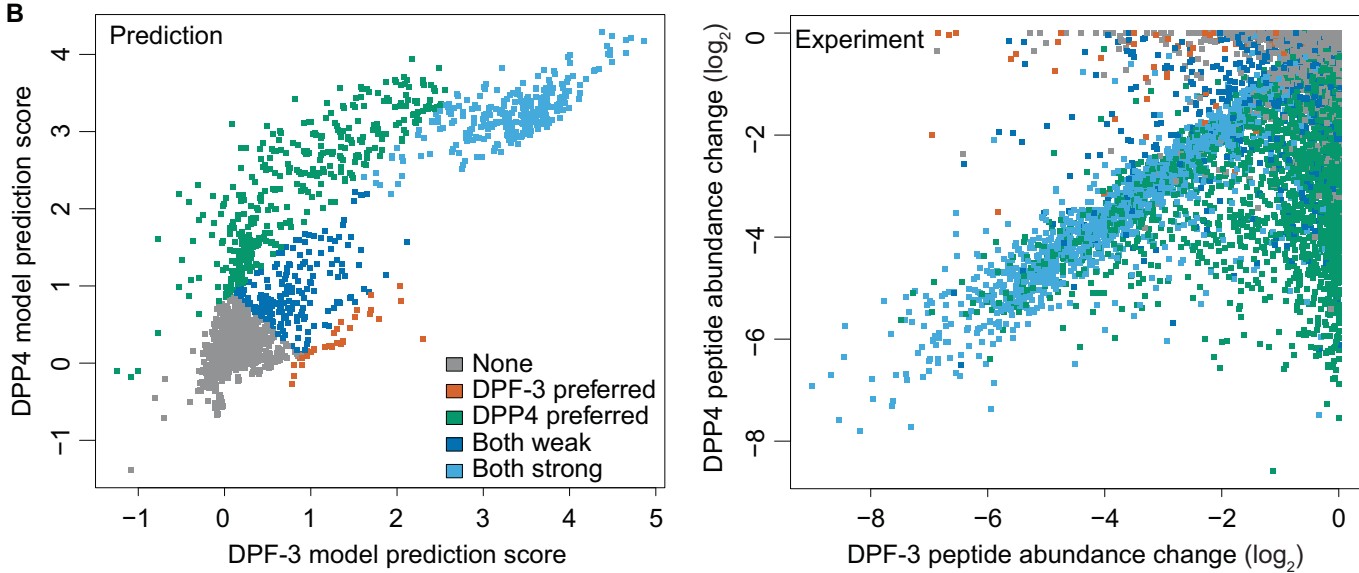

**Figure 6. Substrate preferences of DPF-3 and DPP4 differ.**

(A) Comparison of models generated from an experiment using recombinant DPF-3 (*x* axis) and an independent experiment either using DPF-3 (*y* axis, left) or DPP4 (*y* axis, right). Each dot represents one model parameter. (B) Left panel: Scatter plot showing model prediction scores of all experimentally detected tryptic peptides for DPF-3 (*x* axis) versus DPP4 (*y* axis). The colors denote a visual classification of the peptides as indicated. Right panel: Scatter plot showing peptide abundance changes from the experiments. The colors denote the classification derived from the left panel. Source data are available online for this figure.

appropriate choice of background is indeed a major challenge when analyzing semi-tryptic (product) peptides.

Although qPISA is thus suitable for biased inputs, we point out that certain peptides may be excluded from analysis, e.g., in the current implementation with tryptic digests, substrates containing arginine or lysine in P1 will be depleted. However, digestion with other proteases such as GluC, Chymotrypsin, AspN, or elastase (Dau et al, 2020) can be utilized to ensure sufficient diversity as previously achieved for PICS (Schilling and Overall, 2008). Different sources of biological material

can also be used for lysate preparation, allowing use of lysates from the cell type, tissue or organism for which a user may wish to study the function of their protease of interest. Although not required to establish the general substrate specificity of a protease under investigation, this may facilitate the identification of individual, biologically relevant substrates.

We consider our approach particularly applicable and beneficial for exopeptidases. These account for about 10% (71/703) of human proteases (Rawlings et al, 2018) and their inability to cleave

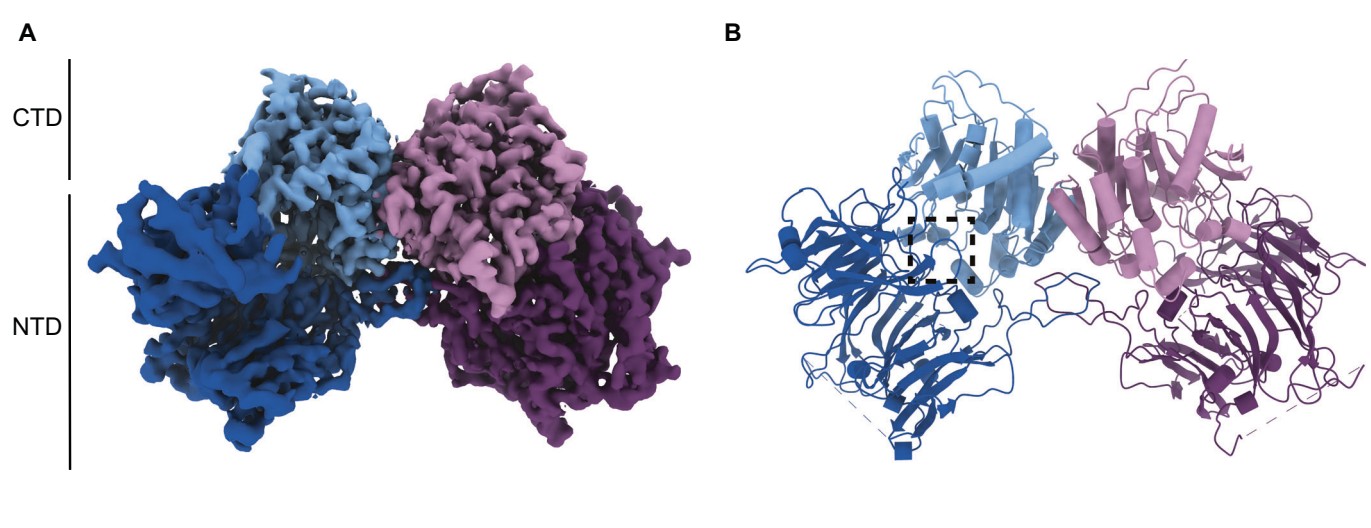

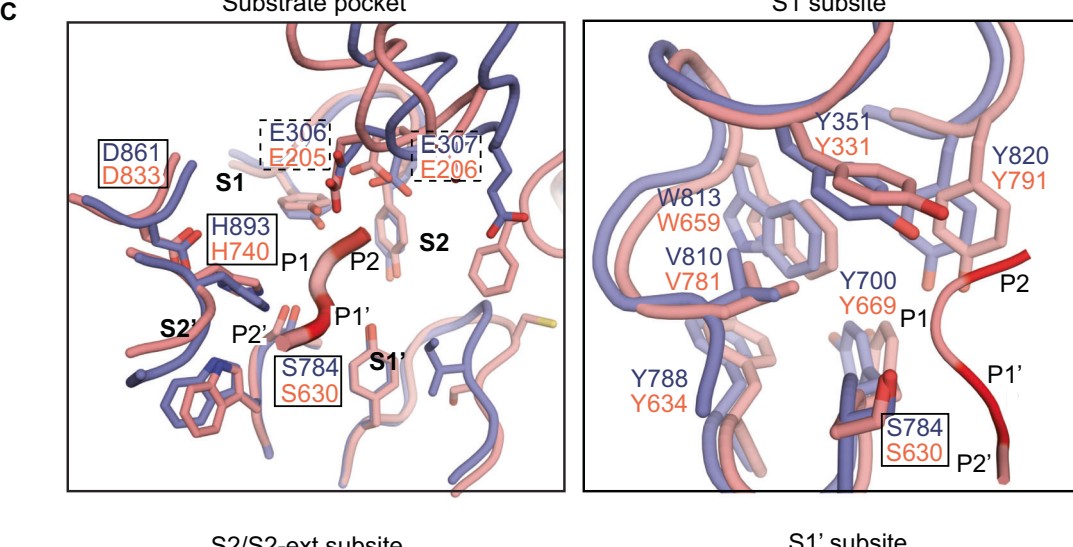

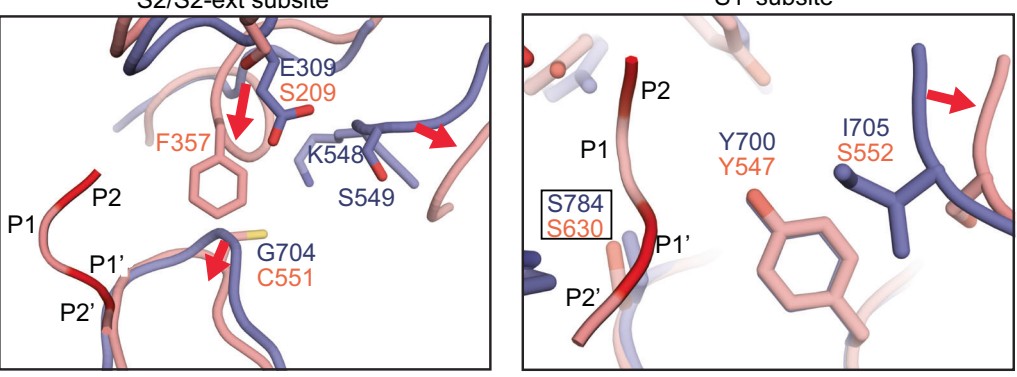

**Figure 7.  Cryo-EM structure of DPF-3 and its comparison with hDPP4.**

(A) Cryo-EM structure of dimeric DPF-4 at 2.6 Å. NTD N-terminal β-propeller domain (dark shade), CTD C-terminal α/β-hydrolase domain (light shade). (B) Atomic model derived from the cryo-EM map shown in (A). The substrate pocket is boxed. (C) Comparison of DPF-3 substrate pocket with DPP4. DPF-3 (blue) superposed on DPP4 (red) in complex with a substrate (red ribbon, PDB 1R9N). The substrate pocket and the S1, S2, and S1' subsites are shown. Substrate residue positions are indicated with P. Residues of the catalytic triad are boxed. The double glutamate motif is highlighted with dashed boxes. Significant backbone shifts are indicated with red arrows.

internally in proteins has made it challenging to study them with techniques such as synthetic peptide libraries (because each peptide encodes only one potential cleavage site) (O'Donoghue et al, 2012) or non-mass spectrometry-based large-scale methods such as phage-display that exploit endoproteolytic cleavage (Matthews and Wells, 1993). For exopeptidases, our analysis can exploit the fact that the cleavage site can be anchored to the protein terminus when analyzing substrates. Nonetheless, we propose that it can also be extended to endopeptidases. In this case, the identification of matching pairs (or trios) of changing substrates and products are used to determine cleavage positions, whereas the accurate quantification of substrate level changes permits a detailed analysis of sequence specificity.

## Substrate specificity of DPP-IV family proteases

Although initially identified through its ability to cleave substrates containing P1(Pro) as reflected in its earlier names glycyl-prolyl-β-naphthylamidase, post-proline dipeptidyl aminopeptidase, and X-Pro dipeptidyl aminopeptidase, it is meanwhile accepted that DPP4 cleaves substrates after residues other than proline (Lambeir et al, 2003). Nonetheless, and given that a preference for P1(Pro) could be well explained by available structural data, we were surprised by the similar level of activity DPP4 achieved on substrates containing P1(Ala). This is unlikely to be an artifact of qPISA since among the five most short-lived substrates whose half-lives were examined by Keane et al (Keane et al, 2011), two carried P1(Ala), including the least stable substrate, GRF-amide.

Despite a major contribution from P1, its identity does not fully specify activity of DPP4. Whether and to what extent other positions contribute has remained largely unknown, in part because most studies were limited to a relatively small number of substrates, and technological differences prevented comparisons across studies (Lambeir et al, 2003). In our analysis, ~40% of variance in the data remained unaccounted in a model that focused exclusively on P1. Importantly, isolated consideration of additional substrate positions provided little further information, and only the combination of residues led to a substantial increase in variance explained. In other words, the impact of the identity of P2 or P1' (on which we focused), can only be understood in the context of the identity of P1. We propose that this context dependency prevented the previous, smaller studies from reaching a consensus in understanding the relevance of these residues.

Given the architecture of the DPP4 active site, cooperative interactions appear best explained by indirect effects of P2 and P1' on P1. For example, side chains packing in a specific way against residues in the S2 and S1' subsites could influence the positioning or backbone geometry, thus aiding in or competing with hydrolysis-competent engagement of the scissile bond. Since proline in P1 on the other hand has a strong positioning effect for the P2 residue due to its constrained backbone geometry and its superior accommodation in the S1 subsite, it may be more compatible than any other residue with the negative effects of certain side chains that are not well tolerated in the S2 subsite. Further changes in the S2 and S1' subsites of DPF-3 relative to DPP4 may then explain why combinatorial effects seem to be even more extensive for DPF-3.

Coverage by our data permitted a near complete survey of the 1200 ($3 \times 20^2$) dipeptide combinations involving P2:P1, P1:P1', and

P2:P1', and these combinations were sufficient to explain most of the experimentally observed substrate specificities. Yet, ~30% of the variance in the data remained unexplained. Conceivably, some of these effects may be explained by peptide structures (Robertson et al, 2016) or post-translational modifications (Mary et al, 2022) that could limit active site accessibility. It also seems likely that more complex interactions among active subsites may exist, which we cannot explore systematically with the current dataset, where we detect for instance 1577 of the possible 8000 tripeptide combinations at sufficient coverage (10 detection events per peptide). Future developments in mass spectrometry and analysis, along with additional adaptations and improvements in sample preparation, can be expected to further expand the utility of the approach to more complex combinations.

## Quantitative substrate analysis provides a basis for protein engineering

Technological advances in protein purification and synthesis have greatly accelerated the development of therapeutic peptide drugs over the past few decades, yet stabilization against proteolytic degradation has remained a bottleneck for successful in vivo application (Wang et al, 2022). Chemical modification can achieve efficient peptide stabilization, but often comes with a trade-off of reduced activity on target (Wang et al, 2022). Our proof-of-principle experiments suggest that qPISA analysis can extend the peptide engineering space, thereby providing further options for the design of stabilized and biologically active peptides.

We note that such engineering efforts are not necessarily limited to peptide stabilization. For instance, it may be beneficial to render specific peptides sensitive against particular proteases to refine or constrain their spatial and temporal activity patterns or modulate, through processing, the type of activity. In the future, through the application of qPISA to other proteases, it may be possible to obtain a compendium of relevant cleavage sites that can be used to engineer synthetic substrates de novo, furthering exploitation of protease-mediated processing for synthetic biology.

## Methods

**Reagents and tools table**

| Reagent/resource | Reference or source | Identifier or catalog number |
|---|---|---|
| **Experimental models** | | |
| **Recombinant DNA** | | |
| **Antibodies** | | |
| **Oligonucleotides and other sequence-based reagents** | | |
| PCR primers | This study | Dataset EV8 |
| **Chemicals, enzymes, and other reagents** | | |
| HeLa Protein Digest Standard | Thermo Fisher Scientific | Catalog# 88329 |
| DPP4 | Abcam | Catalog# ab79138 |
| Q5 Hot Start High-Fidelity 2x Master Mix | NEB | Catalog# E0554S |

| Reagent/resource | Reference or source | Identifier or catalog number |
|---|---|---|
| Cellfectin II Reagent | Thermo Fisher Scientific | Catalog# 10362100 |
| cOmplete EDTA-free protease inhibitor tablet | Roche | Catalog# 11873580001 |
| **Software** | | |
| ChimeraX | https://www.cgl.ucsf.edu/chimerax/ Pettersen et al, 2021 | |
| Molprobity | http://molprobity.manchester.ac.uk/ Williams et al, 2018 | |
| **Other** | | |
| iST-NHS kit for de-salting | PreOmics | Catalog# P.O.00026 |

## qPISA sample preparation and mass spectrometry analysis to determine DPF-3 and DPP4 substrate motifs

The tryptic Pierce HeLa Protein Digest Standard (Thermo Fisher Scientific, 88329) was used as a diverse source of tryptic peptides and incubated with the enzyme of interest in HEPES buffer (HEPES pH 7.4 20 mM, NaCl 150 mM, 2 mM TCEP). Assays were performed in 100 µl volume comprising 10 µg of HeLa protein digest standard and 5 µg of recombinant DPF-3 (see below) or 0.5 µg of DPP4 (Abcam ab79138). The reactions were incubated at 21 °C for 4 h followed by heat inactivation at 95 °C for 3 min. TMT labeling and de-salting was done using the PreOmics iST-NHS kit (Cat # P.O.00026) as described in (Challa et al, 2021). The digested samples were injected, loaded, desalted, and then separated on a 50 cm µPAC C18 HPLC column (Pharmafluidics) connected to a modified Digital PicoView nano-source (New Objective). The following chromatography method was used: 0.1% formic acid (buffer A), 0.1% formic acid in acetonitrile (buffer B), gradient 100 min in total, flow rate 800 nl/min (up to 25 min), then 500 nl/min (25–100 min), mobile phase compositions in % B: 0–5 min 2–7%, 5–25 min 3–6%, 25–30 min 6–8%, 30–70 min 8–20%, 70–88 min 20–32%, 88–89 min 32–95%, 98–96 min 95%, 96–97 min 95–2%, 97–100 min 2%. An Orbitrap Fusion Lumos Tribrid mass spectrometer was operated in a data-dependent mode to quantify TMT reporter ions using synchronous precursor selection-based MS3 fragmentation, as described in (McAlister et al, 2014). Briefly, every 3 s, the most intense precursor ions from Orbitrap survey scans (MS1) were selected for collision-induced dissociation fragmentation with a fixed collision energy set to 32%. MS2 CID spectra were generated by the instrument's ion-trap analyzer from which the ten most abundant notches were selected for MS3 scans. MS3 spectra were recorded using the Orbitrap analyzer at a resolution of 50,000.

MS2 fragment ion spectra were searched with the Sequest HT search engine against the Human SwissProt protein fasta database (downloaded on November 20, 2020). A maximum of two missed cleavages was tolerated for trypsin (semi) enzymatic digestion specificity. Fixed peptide modifications were set for TMTpro/ + 304.207 Da (Any N-Terminus), TMTpro / + 304.207 Da (K), Carbamidomethyl/ + 57.021 Da (C). Variable peptide modifications

were allowed for Oxidation/ + 15.995 Da (M), Acetyl-noTMTpro/- 262.197 Da (N-Terminus). Peptide-to-spectrum matches (PSMs) were validated using the target-decoy search strategy (Elias and Gygi, 2007) and Percolator (Kall et al, 2007) with a strict confidence threshold of 0.01, and a relaxed confidence threshold of 0.05. Unique+razor peptides were used for MS3-based TMTpro reporter-ion quantification, considering Quan Value Correction for isotopic impurities of the TMTpro reagents (Lot# VB294905 and VJ313476), requiring min. PSP mass matches of 65%.

The PeptideGroups table was imported into the peptide workflow in the in-house developed einprot R package version 0.7.0 (Soneson et al, 2023), [https://github.com/fmicompbio/einprot] to undergo $\log_2$ transformation, sample normalization using the center.median approach, and imputation via the MinProb method.

The normalized peptide data was loaded into R as an einprot sce object using the SingleCellExperiment R package (Amezquita et al, 2020), from which peptides with ambiguous N-terminal flanks or with very low detection levels were removed. We used a $\log_2$ intensity threshold of $-4.7$ for the main experiment (with DPP4 and DPF-3) and a threshold of $-4.3$ for the DPF-3 only experiment (after inspecting the respective intensity distributions). For product analysis, peptides mapping to multiple positions in the genome, the first mapping position was utilized arbitrarily to define the matching substrate. Differences in acetylation and/or oxidation levels meant that some peptides had more than one entry. To proceed with distinctive peptide sequences for subsequent motif analyses, only the most abundant form of redundant peptides were retained. Data can be downloaded from the Pride repository (https://www.ebi.ac.uk/pride/) with the accession number PXD042089.

## qPISA data analysis and sequence logo generation

To identify predictive features of cleavage efficiency, we fitted the changes in all tryptic peptides detected by qPISA to increasingly complex linear models, as detailed in the Results section. Linear models are a class of simple but highly interpretable continuous predictors. Given a set of independent variables as input, a linear model calculates an output based on a weighted sum of the independent variables plus an intercept. The weights and the intercept represent the model parameters and are fit such that the correlation between the output of the model and the observed data is maximized. Categorical variables such as amino acid sequence (20 possible values) can be incorporated into a linear model by converting the categorical variable with 20 possible states into $20 - 1 = 19$ dummy variables. The "left out" amino acid is represented by zeros in all of the 19 dummy variables. Multiple amino acid positions can be incorporated by concatenating blocks of dummy variables. Effects of cooperativity between pairs of amino acid positions can be incorporated by including interaction terms. The performance of a linear model is typically quantified by $R^2$ which is the amount of variability in the data that it can explain. Here we use a slight variation, namely the adjusted $R^2$. This measure attempts to account for the complexity of the model and allows for a more fair comparison between models that contain different numbers of parameters.

Once we had identified and fit the final model that included parameters for the position effects of P2, P1, and P1' as well as interaction effects of P2:P1 and P1:P1', we extracted all the coefficients and rearranged them into derived terms that allowed

for simpler interpretability and visualization (i.e., in sequence logos). To do so, we first added back the left-out amino acids at all positions and interaction terms at a value of zero and mean normalized all the coefficients. Secondly, we moved the contributions of P2 and P1' into the interaction terms of P2:P1 and P1:P1' which reduced the total number of terms that need to be added up for every prediction from 6 (Intercept, P2,P1,P1',P2:P1,P1:P1') to only 4 (Intercept, P1,P2:P1,P1:P1'). Third, we moved the intercept term and all the shifts caused by the various mean normalization steps into the coefficients of P1. This resulted in a total of 3 terms that need to be added for a given prediction (P1,P2:P1,P1:P1'). Note that two substrates examined by Keane et al (Keane et al, 2011) were annotated to undergo multiple, successive cleavages; hence, these were excluded from our analysis that examined the correlation between model scores and half-lives since we could not assign a unique experimental half-life to a given tripeptide sequence combination.

## Product-based linear modeling

We used the same overall modeling approach as for the tryptic peptides, attempting to predict the $\log_2$ fold change of the product peptides as opposed to the tryptic peptides. We selected as foreground the product (semi-tryptic) peptides that showed a $\log_2$ fold increase of at least one in the experiment and extended them by two amino acids at the N-terminus. The choice of a suitable background is of critical importance and should ideally account for the non-uniform distribution of amino acids in the HeLa cell proteome, the loss in K/R due to trypsin pre-digestion and the detection biases of the mass spectrometer. For example, a selection of random peptides from the proteome would capture the non-uniform distribution of amino acids but would fail to capture the loss in K/R and the detection biases of the mass spectrometer. Hence, we chose to use tryptic peptides that did not change in intensity ($\log_2$ fold decrease of less than 1) upon incubation as a background. These peptides reflect the skew in amino acid composition in the HeLa proteome, exhibit K/R depletion and account for the detecton bias of the spectrometer. To enable linear modeling, we concatenated the foreground and background peptide data into a single vector using the observed $\log_2$ fold change of the semi-tryptic peptides and setting the $\log_2$ fold change of the tryptic backround peptides to 0.

## Modeling of GLP-1 variant peptide with its receptor

Modeling of a GLP-1 peptide with histidine replaced by aspartate at position 7 was carried out in ChimeraX using the 'Rotamers' tool. As template, PDB ID $6 \times 18$ (Zhang et al, 2021) was used. Clashes for all possible aspartate rotamers (assuming a fixed backbone) were calculated with standard overlap settings (VDW overlap $>= 0.6$ Å after subtracting 0.4 Å for H-bonding). Two rotamers similar to the histidine conformation showed no clashes, of which one was selected for a representative figure (Fig. 4C).

## Production of recombinant DPF-3

C. elegans wild-type or S784A mutated DPF-3 cDNA was generated from RNA extracted from N2 (wt) and dpf-3(xe71[S784A]) mutant animals, respectively. The cDNA was used to amplify and clone the

DPF-3 into the pAC8 vector using Gibson assembly. DNA oligonucleotides used in this study are listed in Dataset EV8. Strep tag was inserted just before the translation stop codon of DPF-3. Truncated (amino acids 216–252 are replaced by 2× Gly Ser) DPF-3 constructs were produced using Q5 Hot Start High-Fidelity 2× Master Mix (New England Biolabs, NEB # E0554S) followed by the KLD reaction according to the manufacturer's instructions. The proteins were expressed and purified exactly as described (Kassube and Thoma, 2020) except that the Trichoplusia ni High-Five insect cells expressing the desired proteins were grown at 16 °C instead of 27 °C. Briefly, Spodoptera frugiperda (Sf9) insect cells grown in EX-CELL 420 medium (Sigma) were used to produce Baculoviruses through recombination by cotransfection of the pAC-derived vector having the gene of interest and viral DNA using Cellfectin II Reagent (Catalog number: 10362100, Thermo Fisher Scientific). The viruses were amplified till Passage 3 (P3) in Sf9 insect cells and the Trichoplusia ni High-Five insect cells were used for large-scale protein expression and purification. Post infection, the cells were allowed to grow at 16 °C for 5 days at 120 r.p.m, collected and lysed by sonication in buffer containing 50 mM Tris pH 7.5, 150 mM NaCl, 1 mM TCEP, 2 mM PMSF and cOmplete EDTA-free protease inhibitor tablets (Roche, 1 tablet per 50 ml). The lysate was clarified by ultracentrifugation for 30 min at $40{,}000 \times g$. Miracloth (Merck) was used to filter the supernatant before loading onto the affinity column.

## In vitro protease assay by mass spectrometry

Synthetic peptides representing either wild-type or mutated GLP-1 were purchased from Thermo Fisher Scientific and their cleavage by DPP4 was measured over time. Briefly, the lyophilized peptides were dissolved in TBS buffer supplemented with 2 mM TCEP to obtain a stock solution with a final concentration of 0.1 nmol/μl. 1ug of DPP4 and each peptide at a final concentration of 1 pmol/μl were used in reactions at a final volume of 100 μl. To distinguish the products, the four peptides were pooled into two groups. One group contained GLP-1 wt (HAE (HAEGTFTSDVSR)) & GLP-1 H1D, E3P (DAP (DAPGTFTSDVSR)) and the other group GLP-1 H1D (DAE (DAEGTFTSDVSR)) & GLP-1 E3P (HAP (HAPGTFTSDVSR)). The assay was performed at 21 °C, aliquots (20 μl) were taken out at the indicated times and mixed with 180 μl of 0.1% trifluoroacetic acid, 2 mM TCEP, 2% acetonitrile in water to stop the reaction.

The peptides were analyzed by capillary liquid chromatography tandem mass spectrometry (LC-MS) using an Orbitrap Fusion LUMOS, a VanquishNeo-nLC, and a 75umx15cm EasyC18 column kept at 45 °C and mounted on a easy source (Thermo Fisher Scientific). Peptides were loaded on a 0.3 × 5 mm C18 trap using a back flash method for elution. Peptides were separated with a gradient of 0–0.2 min 2–6%B in A, 0.2–4.2 min 6–21%, 4.2–6.2 min 21–30%, 6.2–6.7 min 30–36%, 6.7–7.2 min 36–45%, 7.2–7.5 min 45%-100%, 7.5-12 min 100%. A: 0.1% FA in H₂O; B: 0.1% FA, 80% MeCN in H₂O, at room temperature and the flow rate during the gradient was 200 nl/min and for the last 4.4 min 350 nl/min. The data were acquired using 120,000 resolution for the peptide measurements in the Orbitrap and a top T (1.5 s) method with HCD fragmentation for each precursor and fragment measurement in the ion trap following the manufacturer guidelines (Thermo Scientific). MS1 signals were quantified using Skyline 4.1 (MacLean et al, 2010) to generate the results shown in Figs. 4A,B and EV2.

## Cryo-EM sample preparation

In total, 3.5 μl of full-length DPF-3 sample (~0.3 mg/ml and 2.5% glycerol) was applied to Quantifoil holey carbon grids (R 1.2/1.3 200-mesh, Quantifoil Micro Tools). Glow discharging was carried out in a Solarus plasma cleaner (Gatan) for 15 s in a $H_2/O_2$ environment. Grids were blotted for 3–4 s at 4 °C at 100% humidity in a Vitrobot Mark IV (FEI, Hillsboro, OR, USA), and then immediately plunged into liquid ethane.

## Cryo-EM data collection

Data were collected automatically with EPU (Thermo Fisher Scientific) on a Cs-corrected (CEOS GmbH, Heidelberg, Germany) Titan Krios (Thermo Fisher Scientific) electron microscope at 300 keV. A total of 2391 movies were recorded using the Falcon4 direct electron detector (Thermo Fisher Scientific). The acquisition was performed at a magnification of 75,000× yielding a pixel size of 0.845 Å at the specimen level. The dose rate was set to 9.5 $e^-/\text{Å}/s$ and the exposure time was adjusted to account for an accumulated dose of 50 $e^-/\text{Å}^2$. The EER frames were grouped to obtain 50 fractions. The targeted defocus values ranged from −0.8 to −2.5 μm.

## Cryo-EM image processing

Real-time evaluation along with acquisition with EPU (Thermo Fisher Scientific) was performed with *CryoFLARE* (Schenk et al, 2020). Drift correction was performed with the *Relion 3* motioncorr implementation (Zivanov et al, 2018), where a motion-corrected sum of all frames was generated with and without applying a dose weighting scheme. The CTF was fitted using *GCTF* (Zhang, 2016). All the motion-corrected micrographs that showed a CTF estimation better than 4 Å resolution were imported into cryoSPARCv3 (Punjani et al, 2017) for further processing. After patch CTF and blob picker particle picking, 3 rounds of 2d classification were used to obtain a subset of 375,774 particles that were imported into Relion for further processing. A random subset including 140,000 particles were used to generate two ab initio models in cryoSPARCv3, one of each was used as initial model in Relion.

A combination of 3D classification, 3D refinement, particle polishing and CTF refinement in Relion (Fig. EV4) led to a DPF-3 map at 2.6 Å resolution. The resolution values reported for all reconstructions are based on the gold-standard Fourier shell correlation curve (FSC) at 0.143 criterion (Rosenthal and Henderson, 2003; Scheres, 2012) and all the related FSC curves are corrected for the effects of soft masks using high-resolution noise substitution (Chen et al, 2013). All local resolutions were estimated with *MonoRes* (XMIPP) (de la Rosa-Trevin et al, 2013).

## Cryo-EM model building, refinement, and validation

As an initial model, a homology model for DPF-3 was obtained from Swissprot and docked into the structure using CHIMERAX (Pettersen et al, 2021) 'fit-in-map' tool followed by flexible-fitting using ISOLDE (Croll, 2018). The model was further refined by iterative rounds of manual rebuilding in COOT (Emsley et al, 2010) and ISOLDE, followed by minimization using the ROSETTA FastRelax protocol in combination with density scoring

(Wang et al, 2016). For the latter step, an in-house pipeline was used to automate the ROSETTA protocol (Georg Kempf, 2021: https://github.com/fmi-basel/RosEM). Refinement in ROSETTA was done against the first half-map, and the map-model FSC was compared to the map-model FSC of the second half-map to test for overfitting. During the course of this study, ALPHAFOLD 2.0 (Jumper et al, 2021) became available and the predicted structure was used for cross-validation of medium-resolution segments. At the final stage, B-factors were fitted using ROSETTA. The structure was validated with MOLBROBITY (Williams et al, 2018), PHENIX (Afonine et al, 2018) and EMRINGER (Barad et al, 2015). Local amplitude scaling was performed with LocScale (Jakobi et al, 2017).

## Data availability

The electron cryo-microscopy map and model coordinates were deposited in the Electron Microscopy Data Bank (EMDB) (accession code: EMD-17582) and in the Protein Data Bank (PDB) (accession code: 8PBA). The mass spectrometry proteomics data have been deposited to the ProteomeXchange Consortium via the PRIDE (Perez-Riverol et al, 2022) partner repository with the dataset identifier PXD042089. This submission also includes two R data files (.rds) that contain metadata as well as the data that were created in the intermediate processing steps. The code to generate the linear models and the einprot package are available from GitHub (https://github.com/fmicompbio/einprot and https://github.com/fmi-basel/ggrossha-ProteaseSpecificity-qPISA). Archives of the einprot versions used here are available from https://doi.org/10.5281/zenodo.8031403 (v0.6.8) and https://doi.org/10.5281/zenodo.8031408 (v0.7.0). Published research reagents from the FMI are shared with the academic community under a Material Transfer Agreement (MTA) having terms and conditions corresponding to those of the UBMTA (Uniform Biological Material Transfer Agreement).

The source data of this paper are collected in the following database record: biostudies:S-SCDT-10_1038-S44320-024-00071-4.

## Peer review information

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

## Acknowledgements

The authors thank Ganesh Pathare for his kind help and suggestions with regard to the production of recombinant protein and Alicia Michael, Joscha Weiss and Georg Petzold for helpful discussions. We thank Kathrin Braun and Thomas Welte for their inputs and stimulating discussions. The authors thank Nicolas Thomä and Guillaume Diss for their input on earlier manuscript versions, and Nicolas Thomä for access to laboratory resources. The authors are grateful to Vytautas Iesmantavicius for help with peptide fractionation and recording of LC-MS data. This work was supported as a part of the NCCR RNA & Disease, a National Centre of Excellence in Research, funded by the Swiss National Science Foundation (grant number 182880), by the Novartis Research Foundation through the Friedrich Miescher Institute for Biomedical Research (to HG) and by the National Science Center (NCN), Poland, SONATA BIS 2021/42/E/NZ1/00336 and OPUS 2022/45/B/NZ2/02183 (to RKG).

## Author contributions

**Rajani Kanth Gudipati**: Conceptualization; Formal analysis; Investigation; Methodology; Writing—original draft. **Dimos Gaidatzis**: Conceptualization; Software; Formal analysis; Investigation; Visualization; Methodology; Writing—original draft. **Jan Seebacher**: Formal analysis; Investigation; Writing—review and editing. **Sandra Muehlhaeusser**: Resources. **Georg Kempf**: Validation; Visualization; Writing—original draft. **Simone Cavadini**: Formal analysis; Visualization. **Daniel Hess**: Formal analysis; Investigation. **Charlotte Soneson**: Software. **Helge Großhans**: Conceptualization; Supervision; Funding acquisition; Writing—original draft; Project administration.

Source data underlying figure panels in this paper may have individual authorship assigned. Where available, figure panel/source data authorship is listed in the following database record: biostudies:S-SCDT-10_1038-S44320-024-00071-4.

## Disclosure and competing interests statement

The authors declare no competing interests.

# Expanded View Figures

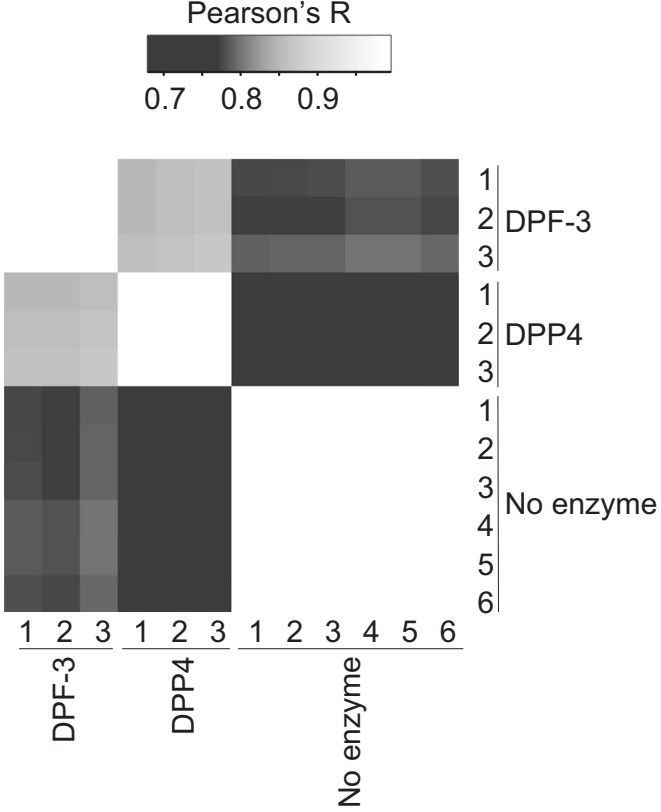

**Figure EV1.  Determination of DPP4 cleavage motif.**

Pairwise correlation plot of identified peptides for each experimental condition and sample as indicated, showing Pearson's correlation coefficient. Label numbers indicate technical replicates. In subsequent quantitative analyses, three "No enzyme" control replicates were used for comparison to DPF-3-digested samples, the other three for comparison to DPP4-digested samples.

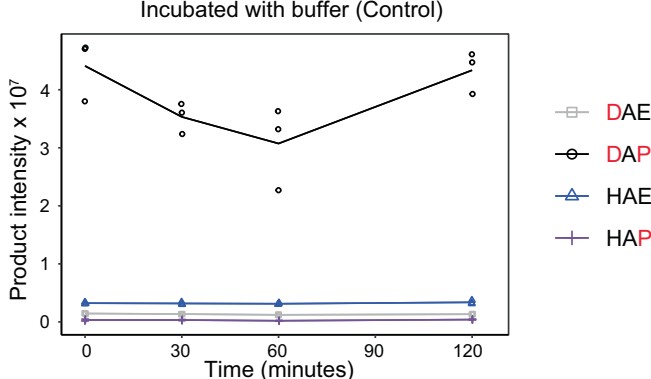

**Figure EV2.   Products are detectable in the synthetic peptide cleavage assay.**

Zoom-in on the panel from Fig. 4B showing product peptide intensities over time following incubation of a GLP-1$_{(7-36)}$-derived peptide and its variants in buffer only (negative control). Time-invariant trace levels (<1% of signal detected in the presence of enzyme) of the expected product peptides are detected as contaminants of the peptide substrates, confirming that all hypothetical products are detectable in the assay.

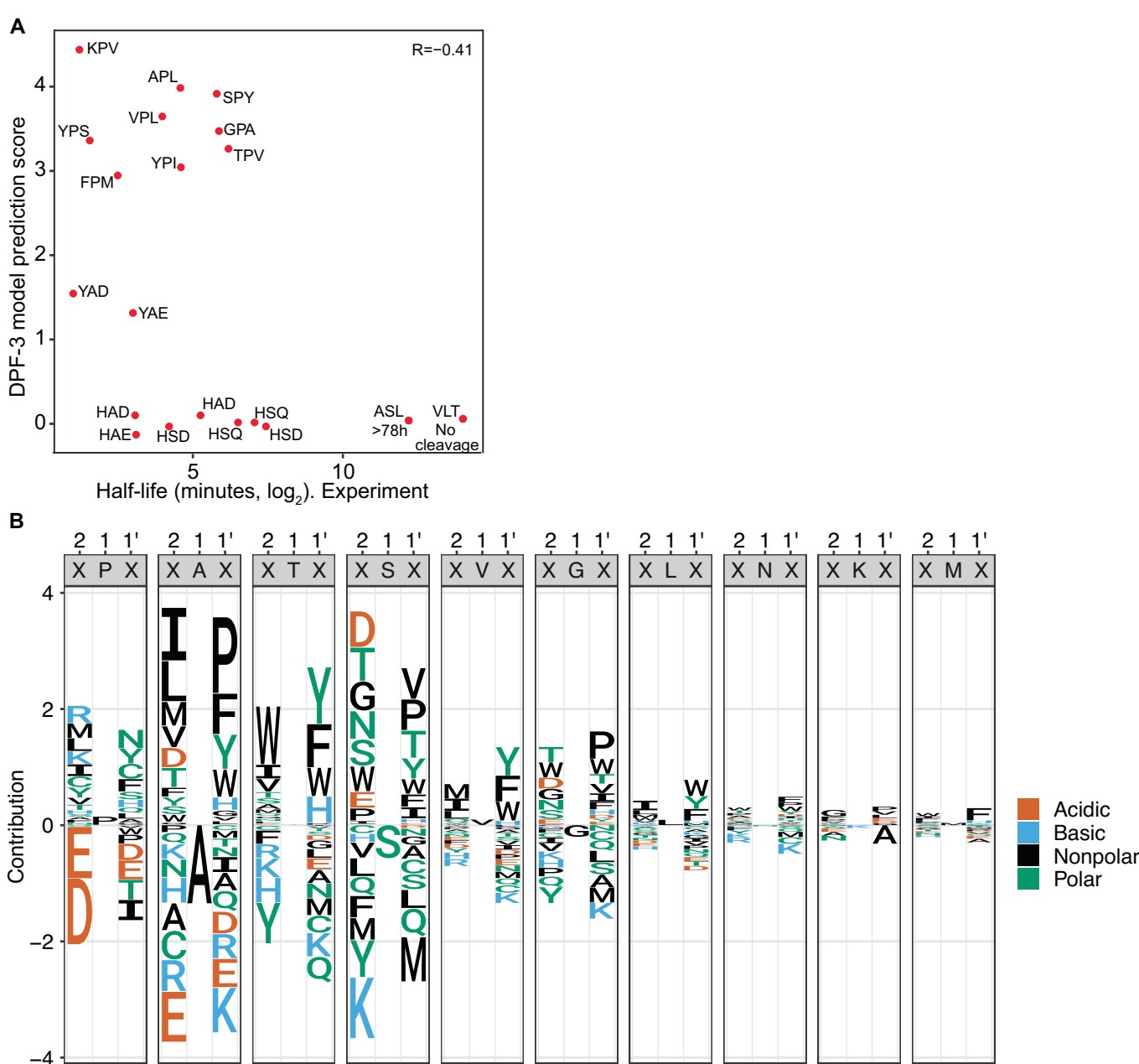

**Figure EV3. Determination of the DPF-3 cleavage motif.**

(**A**) Scatter plot of experimentally determined peptide half-lives upon incubation with DPP4 (from (Keane et al, 2011)) vs. DPF-3 qPISA model substrate score. Each dot is a unique peptide whose first three amino acids are indicated. (**B**) Sequence logo showing the differential cleavage activity (DPP4 subtracted from DPF-3) when the P1 position is occupied by a given amino acid in relation to different amino acids occupying P2 and P1' position.

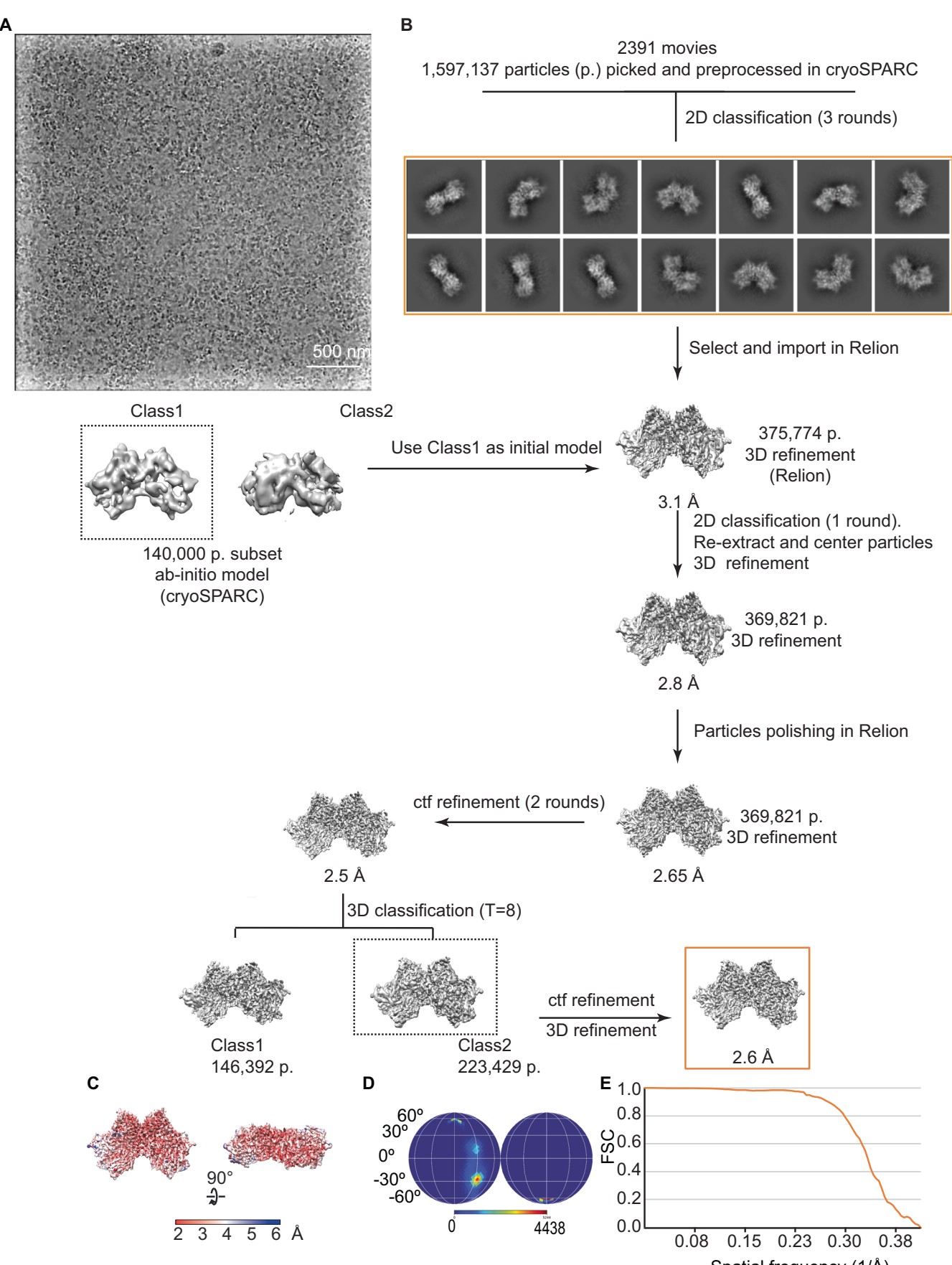

**Figure EV4. Classification and refinement workflow for the DPF-3 homodimer complex.**

(A) Representative cryo-EM micrographs denoised with Topaz, see methods. (B) The movies were imported in cryoSPARC for initial data processing including 2D classification (representative 2D class averages are shown inside an orange frame). The best particles were imported in Relion. Ab initio model generation was performed in cryoSPARC, while 3D classification and refinement were performed in Relion. The final model includes 223,429 particles. The boxes defined by a dashed line indicate the good models and set of particles used for the following step in the data processing workflow. (C) Local resolution filtered map (MonoRes). (D) Angular distribution for the particles leading to the final EM map. (E) Gold-standard Fourier shell correlation curve.

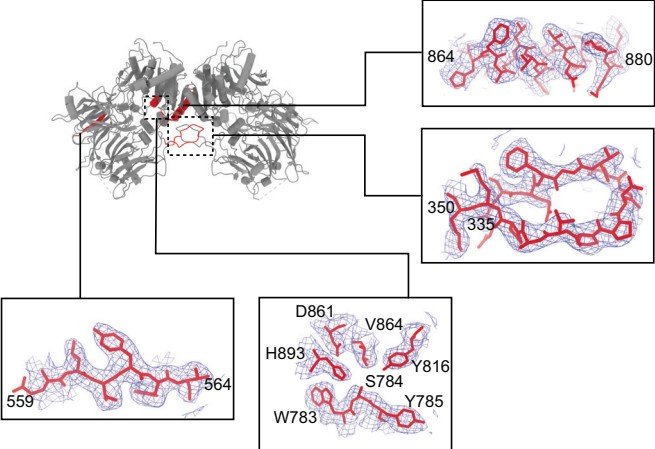

**Figure EV5.   Model map of DPF-3 dimer.**

Representative model-map overlays (insets) for regions colored in red in the top left panel. Numbers correspond to sequence positions of residues. The map was sharpened with LocScale and the contour level is 0.192.

