## [Peer Review File · Molecular Systems Biology]

Deep quantification of substrate turnover defines protease subsite cooperativity

Rajani Gudipati, Dimos Gaidatzis, Jan Seebacher, Sandra Muehlhaeusser, Georg Kempf, Simone Cavadini, Daniel Hess, Charlotte Sonesson, and Helge Großhans

Corresponding author(s): Helge Großhans (helge.grosshans@fmi.ch)

Review Timeline:

Submission Date:	17th Apr 24
Editorial Decision:	27th May 24
Revision Received:	12th Aug 24
Editorial Decision:	12th Sep 24
Revision Received:	15th Oct 24
Accepted:	18th Oct 24

Editors: Maria Polychronidou and Jingyi Hou

Transaction Report:

27th May 2024

Manuscript Number: MSB-2024-12373

Title: Massively parallel quantification of substrate turnover defines protease subsite cooperativity

Dear Prof. Großhans,

Thank you again for submitting your work to Molecular Systems Biology. We have now heard back from two of the three reviewers who agreed to evaluate your study. Unfortunately, after several reminders we did not receive a report from reviewer #1. In the interest of time, we have decided to proceed with making a decision based on the two available reports. As you will see below the reviewers acknowledge that the presented findings are relevant for protease biology and for potential future applications. However, they raise several concerns, which we would invite you to address in a revision.

I think that the reviewers' recommendations are rather clear and I therefore see no need to repeat all the comments listed below. Of note, the reviewers refer to the need improve the benchmarking and to better demonstrate the advantages of the presented approach compared to existing ones. Moreover, reviewer #2 points out that the data and code should be made available for evaluation.

All issues raised by the reviewers would need to be satisfactorily addressed. As you may already know, our editorial policy allows in principle a single round of major revision. It is therefore essential to provide responses to the reviewers' comments that are as complete as possible. If you have any questions or if you would like to discuss your revision plan with me, please feel free to get in touch.

On a more editorial level, we would ask you to address the following points:

- Please provide a .doc version of the manuscript text (including legends for main Figures and EV Figures) and individual production quality figure files for the main Figures and EV Figures (one file per figure).
- Please include 5 keywords.
- We have replaced Supplementary Information by the Expanded View (EV format). In this case (unless the number of EV figures becomes very large during revision), all additional figures can be provided as EV Figures. Please provide one file per EV Figure. Their legends should be included in the manuscript text. For detailed instructions regarding expanded view please refer to our Author Guidelines: .
- Supplementary Tables should be provided as EV Datasets. Please provide one file per EV Dataset. In each file, a description of the table/dataset should be provided in a separate sheet.
- Please provide a "standfirst text" summarizing the study in one or two sentences (approximately 250 characters), three to four "bullet points" highlighting the main findings and a "synopsis image" (exactly 550px width and max 400px height, jpeg or png format) to highlight the paper on our homepage.
- All Materials and Methods need to be described in the main text. We would encourage you to use 'Structured Methods', our new Materials and Methods format. According to this format, the Material and Methods section should include a Reagents and Tools Table (listing key reagents, experimental models, software and relevant equipment and including their sources and relevant identifiers) followed by a Methods and Protocols section in which we encourage the authors to describe their methods using a step-by-step protocol format with bullet points, to facilitate the adoption of the methodologies across labs. More information on how to adhere to this format as well as downloadable templates (.doc or .xls) for the Reagents and Tools Table can be found in our author guidelines: . An example of a Method paper with Structured Methods can be found here:
- Please include a Data availability section describing how the data and code have been made available. This section needs to be formatted according to the example below:
The datasets and computer code produced in this study are available in the following databases:
 - Chip-Seq data: Gene Expression Omnibus GSE46748 (<https://www.ncbi.nlm.nih.gov/geo/query/acc.cgi?acc=GSE46748>)
 - Modeling computer scripts: GitHub (<https://github.com/SysBioChalmers/GECKO/releases/tag/v1.0>)
 - [data type]: [full name of the resource] [accession number/identifier] ([doi or URL or identifiers.org/DATABASE:ACCESSION])
- For data quantification: please specify the name of the statistical test used to generate error bars and P values, the number (n) of independent experiments (specify technical or biological replicates) underlying each data point and the test used to calculate p-values in each figure legend. The figure legends should contain a basic description of n, P and the test applied. Graphs must include a description of the bars and the error bars (s.d., s.e.m.).

- Please include a "Disclosure & Competing Interests Statement" in the main text.
- The References should be formatted according to the Molecular Systems Biology reference style (i.e., ordered alphabetically and listing the first 10 authors followed by et al).
- When you resubmit your manuscript, please download our CHECKLIST (<https://bit.ly/EMBOPressAuthorChecklist>) and include the completed form in your submission.
Please note that the Author Checklist will be published alongside the paper as part of the transparent process (<https://www.embopress.org/page/journal/17444292/authorguide#transparentprocess>).

If you feel you can satisfactorily deal with these points and those listed by the referees, you may wish to submit a revised version of your manuscript. Please attach a point by point response giving details of the way in which you have handled each of the points raised by the referees. A revised manuscript will be once again subject to review and you probably understand that we can give you no guarantee at this stage that the eventual outcome will be favorable.

Kind regards,

Maria

Maria Polychronidou, PhD
Senior Editor
Molecular Systems Biology

We realize that it is difficult to revise to a specific deadline. In the interest of protecting the conceptual advance provided by the work, we recommend a revision within 3 months (25th Aug 2024). Please discuss the revision progress ahead of this time with the editor if you require more time to complete the revisions. Use the link below to submit your revision:

IMPORTANT:

- Please note that corresponding authors are required to supply an ORCID ID for their name upon submission of a revised manuscript (EMBO Press signed a joint statement to encourage ORCID adoption). (<https://www.embopress.org/page/journal/17444292/authorguide#editorialprocess>)
Currently, our records indicate that the ORCID for your account is 0000-0002-8169-6905.

Please click the link below to modify this ORCID:
Link Not Available

*** PLEASE NOTE *** As part of the EMBO Press transparent editorial process initiative (see our Editorial at <https://dx.doi.org/10.1038/msb.2010.72>), Molecular Systems Biology publishes online a Review Process File with each accepted manuscripts. This file will be published in conjunction with your paper and will include the anonymous referee reports, your point-by-point response and all pertinent correspondence relating to the manuscript. If you do NOT want this File to be published, please inform the editorial office at msb@embo.org within 14 days upon receipt of the present letter.

Reviewer #2:

Summary

- Describe your understanding of the story
- What are the key conclusions: specific findings and concepts
- What were the methodology and model system used in this study

In this manuscript the authors describe a very simple method to analyse the substrate specificity of exopeptidases based on the quantification by MS of tryptic peptides using a HeLa protein tryptic digestion as a substrate. The methodological advances are supported with biochemical and structural data relative to the DPP4 and DPF-3 proteases. This is a neat method that enables researchers a deeper understanding of a protease specificity, even though it is mainly suited to the analysis of exopeptidases.

General remarks

- Are you convinced of the key conclusions?
- Place the work in its context.
- What is the nature of the advance (conceptual, technical, clinical)?
- How significant is the advance compared to previous knowledge?
- What audience will be interested in this study?

The data presented is generally convincing and describes a variation of the Proteomic Identification of Cleavage Site Specificity (PICS) protocol, which essentially consists in making PICS more quantitative by focusing on the peptide substrates, rather than the peptide products of the proteolytic reaction. This is of interest to characterize better the substrate specificity of proteases, which post-translationally modify protein substrates, and can be dysregulated in numerous diseases.

Despite the idea of the method for analysing cleavage products from lysates is not new, the simplification of the PICS protocol by eliminating the need of enrichment for prime-side cleavage product is a good idea. The paper also mentions that by using linear models to fit the variance of the data, the strategy forgoes the requirement of arbitrary thresholds when using isotopic tags (<https://doi.org/10.1074/mcp.M000032-MCP201>) or non-isotopic tags (<https://doi.org/10.1074/mcp.O115.056671>). It is also quite novel that the approach focuses on protease substrates, rather than products regarding protease cleavage specificity. However, it would be advisable to provide a more extensive benchmarking about how this implementation performs as compared with the gold standards in the field. It would be interesting to see how much benefit this "novel" method is compared to a "classic" experiment looking at protease specificity from the view of semi-tryptic products. Similar to what was performed by the Schilling group (<https://doi.org/10.1074/mcp.O115.056671>) or with TMT from the Overall group (<https://doi.org/10.1074/mcp.M000032-MCP201>). For instance, one could look at the fold change of products to generate cleavage specificity using some arbitrary threshold. Using the data that has been already generated, it would be nice to see how different the sequence logos would be when one has several hundreds/thousands of peptides to generate a sequence logo from semi-tryptic peptides, compared to this new novel method which looks at plotting a linear model to explain the variance of 40,000 tryptic substrates.

Major points

- Specific criticisms related to key conclusions
- Specify experiments or analyses required to demonstrate the conclusions
- Motivate your critique with relevant citations and argumentation

- "Several mass spectrometry-based techniques such as [...] can successfully identify protease-mediated cleavage events in native contexts or in in vitro lysates by enriching and detecting cleavage products. Yet, they tend to be limited to a few hundred sites per experiment (Tsiatsiani and Heck 2015) and they suffer from an absolute quantification bias of mass-spectrometry where the exact sequence of a peptide influences its detectability and quantification (Liigand, Kaupmees, and Kruve 2019)." While it is true that a lot of the earlier enrichment methods (PICS, Schilling 2008) resulted in very few N-termini, recent depletion protocols, using cheap reagents, can identify a large number of cleavage products and N-termini even with relatively low sample input (Weng, et al, HUNTER, <https://doi.org/10.1074/mcp.TIR119.001560>). This method should also be cited, together with the two manuscripts I referenced earlier (<https://doi.org/10.1074/mcp.M000032-MCP201>, <https://doi.org/10.1074/mcp.O115.056671>).
- Experimental design: it is not clear whether two TMT6-plex were used in this experiment or simply 12 channels of a TMT pro, as both TMTpro and TMT6plex are mentioned in the methods section. In general, the overall understanding of the method workflow can be improved by making Figure 1, panel B of nicer quality.
- Identification, FDR and differential quantification of the tryptic peptides with and without the protease: In method section it is clear that TMT quantification is used but not how the data is processed afterwards. How is the differential abundance analysis of peptides done and their statistical significance assessed? Since the key point of the method is to do this quantification step right, more descriptions are required to understand the method and guarantee its reproducibility. What is the MinProb method?
- Also here: "For peptides with multiple mapping positions, only the first one was considered. To proceed with distinctive peptide sequences for subsequent motif analyses, only the most abundant isoforms of redundant peptides were retained" it is not clear what the authors mean with 'peptide isoforms here'.
- Can the authors comment about the problem of the expansion of the search space.
 - o They are searching for semi-tryptic peptides but they are not going to use them for their quantification analysis. Is this correct?
 - o Is the increase of the search space affecting in any way the confidence/quality of quantification of their substrate peptides?
- More detail is required regarding how data analysis is done. In the method section the procedure of fitting into linear model is explained from a theoretical point of view. It would be nice if the code that was used to generate the linear models could be added so that the method itself could be more easily taken up by the protease community. At least the linear model method

should be explained in such a way that it can be reproduced. In addition, the generation of sequence logos is often done by readily available tools where peptide intensities are very often simply normalized to the abundance of the proteome. Since the data handling for the visualization of the sequence logos appear to be more sophisticated here, it would be advisable to provide an explanation of the algorithm used, or even better share the code.

- Figure 1c: This is a scatter plot. Why the log₂ intensities on the axis are negative? Is this due to normalization? It is quite confusing to what one normally expects from a peptide abundance scatter plot. This panel could be made clearer.
- There is no project named PXD042089 in the Pride database. This should be made available for the reviewers and later for publication.

Minor points

- Easily addressable points
- Presentation and style
- Trivial mistakes

- The quality and layout of figure 1 could be improved
- I somehow argue against the usage of the wording 'massively parallel' in the title or 'highly parallelized' in the main text. I believe this is a bit exaggerated, when the substrate specificity of only two proteases has been described in this manuscript.
- About experimental design explanation, along the lines of my previous comment. In supplementary figure 1 the schematic provided does not help at all in the definition of biological (?) or technical (?) replicates. The figure legend text is also very poor and not helpful for the figure interpretation.
- This sentence: "this analysis would also compensate for certain compositional biases in the input material". What do you mean with that? Can you quickly elaborate in the text?

Reviewer #3:

Summary

The manuscript by Gudipati et al. explores the effectiveness of a method called Quantitative Protease Specificity Inference from Substrate Analysis (qPISA) in analyzing the activity of human dipeptidyl peptidase 4 (DPP4) and the *C. elegans* protease DPF-3, an orthologue of human DPP8/9. This methodology integrates the high-throughput Proteomic Identification of Cleavage Sites (PICS) with a combinatorial analysis of the interactions between different proteolytic subsite features that promote or inhibit cleavage. Using this approach, they developed a substrate prediction model for the studied proteases and demonstrated its ability to predict their substrate proteins' half-lives. Using this model, they examined DPP4's substrate specificity and demonstrated how to engineer one of its known substrates, GLP-1, a peptide with implications for diabetes and obesity treatment. They performed a similar analysis on *C. elegans* DPF-3, comparing its substrate specificity to that of human DPP4. This comparative analysis reveals both similarities and differences in substrate preferences between the two proteases. They also resolved the structure of DPF-3 using cryo-electron microscopy and compared it with the published structure of human DPP4, highlighting the structural features in the S2 and S1' subsites responsible for differences in cleavage preferences.

General remarks

This is a solid work that includes extensive proteomics, structural, and computational studies. The text is clear and easy to follow, although some experimental details lack clarity (see below). The key conclusions about the cleavage preferences and subsite cooperativity of DPP4-particularly the pronounced preference for P1=Ala-are important. These findings enhance our understanding of this enzyme, which may have future implications given the significance of this protease. However, this work does not represent a conceptual breakthrough, as it builds upon a previously published proteomics method (PICS) and reiterates the concept of subsite cooperativity previously presented (e.g., PMID: 31543345). It should be noted that PICS was The conclusions appear to be well-supported by the data, aligning with and validating existing knowledge and hypotheses regarding DPP specificity, including some unpublished studies. The insights provided will undoubtedly be of interest to the community of protease biology researchers, particularly those focused on dipeptidyl peptidases.

Major points

- *Novelty and concept of qPISA workflow

The simplified PICS protocol utilized in this study has been previously published several times (e.g., PMID: 27122596, PMID: 28315252, PMID: 28315250 with TMT, among others).

The authors emphasize the advantage of using the quantification of PICS/qPISA input peptides compared to the quantification of cleaved peptides in PICS. This approach shifts the entire workflow to rely on the lack of signal (disappearance of the input peptides) rather than the appearance of signal (from the cleaved peptides), which provides direct evidence for the presence of cleaved substrates in the sample.

It is not clear how the authors utilize (if at all) the actual quantification of the different peptides in their workflow.

Can the authors demonstrate the advantages of their approach compared to the commonly used approach in PICS-like experiments, which defines specificity based on the identification and quantification of neo-terminal (non-tryptic) peptides? They should compare their results to those that will be obtained based on the non-tryptic peptides. Additionally, how effective is the suggested workflow, which relies on the disappearance of tryptic peptides, for determining the cleavage preferences of peptidases with broader specificity?

***Technical aspects in the used workflow**

The peptide identification list from the DPP9 and DPF-3 experiment includes approximately 18% of peptides (~12,000 out of ~64,000) that were not quantified. This represents a significant portion of unquantified peptides, which is substantially higher than the typical 1-5% usually observed with TMT labeling. This discrepancy raises concerns about the efficiency of the labeling process or the optimization of the MS analysis settings and might contribute to some biases in the results. The authors should perform a database search while setting all TMT modifications as variable, to demonstrate labeling efficiency and explain the reason for the high portion of missed quantifications.

It is unclear how the authors determined the intensity threshold for the different experiments and how they compared the log₂ intensity of the protease-treated samples to the no-enzyme control. Why did they not simply perform a statistical test to compare the ratios of protease to no-enzyme in order to identify significant changes? Additionally, the log₂ intensities of the tryptic peptides in the protease-treated samples, as reported in Suppl. Table 1, shows a broad distribution centered around zero. Can the authors explain the reasons for such a distribution?

***Comparison of DPP4 and DPF-3**

DPF-3 originates from *C. elegans*, and therefore, its target proteome composition and the resulting tryptic peptide library might differ from the human tryptic library used in this study. Can the authors discuss the potential impact of these differences, if any?

***Other factors that might alter specificity**

DPP4 and DPF-3 are relatively large proteins. Could other elements within their structures also contribute to their cleavage specificity shown in this work?

Minor points

* The proteomics data should be reported in detail in supplementary tables, including identified peptide sequences, scores, and quantification. It is recommended to follow the guidelines established in PMID: 23500130 or the HUPO guidelines available at: https://hupo.org/resources/Documents/HPPMSDataGuidelines_3.0.0.pdf

* Please add a supplementary table detailing the full sequences of the 21 peptides and their respective proteins used in Figure 3.

* In Figure 3 (B,C), please include the sequences of the (expected) product peptides. Were other products identified? Please provide the tables for peptide identification and quantification as a supplement.

* What is COCharDiC (Page 2)? Could this be a typo for ChFradic? If so a different reference might be required (e.g. PMID: 26010716).

Reviewer #2:

Summary

- Describe your understanding of the story
- What are the key conclusions: specific findings and concepts
- What were the methodology and model system used in this study

In this manuscript the authors describe a very simple method to analyse the substrate specificity of exopeptidases based on the quantification by MS of tryptic peptides using a HeLa protein tryptic digestion as a substrate. The methodological advances are supported with biochemical and structural data relative to the DPP4 and DPF-3 proteases. This is a neat method that enables researchers a deeper understanding of a protease specificity, even though it is mainly suited to the analysis of exopeptidases.

We thank the reviewer for their endorsement of our work and the thoughtful comments that we address in detail below.

General remarks

- Are you convinced of the key conclusions?
- Place the work in its context.
- What is the nature of the advance (conceptual, technical, clinical)?
- How significant is the advance compared to previous knowledge?
- What audience will be interested in this study?

The data presented is generally convincing and describes a variation of the Proteomic Identification of Cleavage Site Specificity (PICS) protocol, which essentially consists in making PICS more quantitative by focusing on the peptide substrates, rather than the peptide products of the proteolytic reaction. This is of interest to characterize better the substrate specificity of proteases, which post-translationally modify protein substrates, and can be dysregulated in numerous diseases.

Despite the idea of the method for analysing cleavage products from lysates is not new, the simplification of the PICS protocol by eliminating the need of enrichment for prime-side cleavage product is a good idea. The paper also mentions that by using linear models to fit the variance of the data, the strategy forgoes the requirement of arbitrary thresholds when using isotopic tags (<https://doi.org/10.1074/mcp.M000032-MCP201>) or non-isotopic tags (<https://doi.org/10.1074/mcp.O115.056671>). It is also quite novel that the approach focuses on protease substrates, rather than products regarding protease cleavage specificity. However, it would be advisable to provide a more extensive benchmarking about how this implementation performs as compared with the gold standards in the field. It would be interesting to see how much benefit this "novel" method is compared to a "classic" experiment looking at protease specificity from the view of semi-tryptic products. Similar to what was performed by the Schilling group (<https://doi.org/10.1074/mcp.O115.056671>) or with TMT from the Overall group

(<https://doi.org/10.1074/mcp.M000032-MCP201>). For instance, one could look at the fold change of products to generate cleavage specificity using some arbitrary threshold. Using the data that has been already generated, it would be nice to see how different the sequence logos would be when one has several hundreds/thousands of peptides to generate a sequence logo from semi-tryptic peptides, compared to this new novel method which looks at plotting a linear model to explain the variance of 40,000 tryptic substrates.

We thank the reviewer for the suggestion to provide more extensive benchmarking. We point out that our approach involves two innovations: Analysis of substrate instead of product peptides, and quantitative modeling that, as the reviewer rightly notes above, foregoes the need for setting arbitrary cut-offs to define substrates. It is the combination of the two that makes the approach powerful and allows us to capture the interaction terms (as illustrated in Fig. 2B vs. 2C for the substrates). We show sequence logos for tryptic and semi-tryptic peptides in the Reviewer Figure 1 below, each of which does a good job in identifying suitable amino acids in P1 but neither of which can uncover protease subsite cooperativity, a key objective of our study. Hence, we went ahead and quantitatively modeled the product peptide changes including interaction terms, following the same approach as for the substrates. As detailed in the new section “Substrate versus product: a comparison” with the new Fig. 3, we find that this analysis yields qualitatively similar results to the tryptic peptide analysis but performs quantitatively less well. It also requires more assumptions regarding background choice. (See also our comment further down concerning how qPISA can compensate for biases in sample composition.) We conclude that qPISA outperforms product-based analysis.

Reviewer Figure 1: Sequence logos using A) tryptic and B) semi-tryptic peptides. Panel A is identical to Fig. 2B in the manuscript. Panel B uses a total of 41,571 peptides, 3,600 product peptides as the foreground and 37,971 as the background, selected as detailed in the Methods section under “Product based linear modelling”.

Major points

- Specific criticisms related to key conclusions
- Specify experiments or analyses required to demonstrate the conclusions

-Motivate your critique with relevant citations and argumentation

- "Several mass spectrometry-based techniques such as [...] can successfully identify protease-mediated cleavage events in native contexts or in in vitro lysates by enriching and detecting cleavage products. Yet, they tend to be limited to a few hundred sites per experiment (Tsiatsiani and Heck 2015) and they suffer from an absolute quantification bias of mass-spectrometry where the exact sequence of a peptide influences its detectability and quantification (Liigand, Kaupmees, and Kruve 2019)."

While it is true that a lot of the earlier enrichment methods (PICS, Schilling 2008) resulted in very few N-termini, recent depletion protocols, using cheap reagents, can identify a large number of cleavage products and N-termini even with relatively low sample input (Weng, et al, HUNTER, <https://doi.org/10.1074/mcp.TIR119.001560>). This method should also be cited, together with the two manuscripts I referenced earlier (<https://doi.org/10.1074/mcp.M000032-MCP201>, <https://doi.org/10.1074/mcp.O115.056671>).

Thank you, we have revised our introduction to include these relevant advances.

- Experimental design: it is not clear whether two TMT6-plex were used in this experiment or simply 12 channels of a TMT pro, as both TMTpro and TMT6plex are mentioned in the methods section. In general, the overall understanding of the method workflow can be improved by making Figure 1, panel B of nicer quality.

We apologize for the confusion, TMTpro was used exclusively. We have corrected the text. We interpret the reviewer's request for a "panel B of nicer quality" to refer to a visually more appealing version, which we attempted to generate through the inclusion of schematic cartoons. We hope that the reviewer finds it to be to their liking.

- Identification, FDR and differential quantification of the tryptic peptides with and without the protease: In method section it is clear that TMT quantification is used but not how the data is processed afterwards. How is the differential abundance analysis of peptides done and their statistical significance assessed? Since the key point of the method is to do this quantification step right, more descriptions are required to understand the method and guarantee its reproducibility. What is the MinProb method?

Concerning the differential abundance analysis, we did not aim to distinguish changing peptides from non-changing peptides in a binary fashion using a statistical test. Instead, we calculated the \log_2 fold change between the buffer control samples and the enzyme-treated samples and devised linear models to predict that quantity. We added an additional description in the manuscript to clarify this point (new text underlined):

"To this end, we devised a linear model to predict the tryptic substrate peptide intensity changes in the experiment from their amino acid sequences. Linear models (described in more detail in Methods) use a set of independent variables as input to calculate an output

such that the correlation between the output of the model and the observed data is maximized. This approach does not require the selection of potential substrates through a statistical test to distinguish changing peptides from non-changing peptides in a binary fashion. Instead, the model directly predicts the \log_2 fold change between the buffer control samples and the enzyme-treated samples. Its performance can be quantified by the R^2 , which is the amount of variability in the data that it can explain. In the following, to allow for a fairer comparison among models of different complexities, we use the adjusted R^2 , which accounts for different numbers of parameters in the models.“

The MinProb method allows imputation of left-censored missing data using a stochastic minimal value approach; it is available in the imputeLCMD package used in einprot (Soneson, et al., 2023. 'einprot: flexible, easy-to-use, reproducible workflows for statistical analysis of quantitative proteomics data', *Journal of Open Source Software*, 8: 5750).

- Also here: "For peptides with multiple mapping positions, only the first one was considered. To proceed with distinctive peptide sequences for subsequent motif analyses, only the most abundant isoforms of redundant peptides were retained" it is not clear what the authors mean with 'peptide isoforms here.

We apologize for the confusion; we were referring to the fact that in our quantification table, a small subset of peptides occurred in duplicates, where the peptides differed e.g. in acetylation or oxidation status. Since a motif analysis relies on distinctive sequences, we retained only the most abundant peptide for the model. We have clarified this in the methods section and replaced the term “isoform”

- Can the authors comment about the problem of the expansion of the search space.

- o They are searching for semi-tryptic peptides but they are not going to use them for their quantification analysis. Is this correct?

- o Is the increase of the search space affecting in any way the confidence/quality of quantification of their substrate peptides?

We decided to increase the search space to include semi-tryptic peptides, as this represents more faithfully the situation in the digested sample, thus reducing the risk of tryptic peptide misidentification. We used the semi-tryptic peptides in Fig. 1C, where our observation of the distinct behaviors of tryptic and semi-tryptic peptides helped to motivate our analysis of tryptic (substrate) peptides. We are now also relying on these data for the benchmarking experiments that the reviewer requested (described above). Nonetheless, it is true that the increase in search space could increase both false positive and false negative identifications. We did not expect a negative effect on the ability to confidently identify N-terminally semi-tryptic peptides as they show the same MS ionization and fragmentation characteristics as fully tryptic peptides (positive charges on

each end). The vast majority of semi-tryptic peptides are N-terminally semi-tryptic. In case of an issue with the confidence of identification, one would expect similar numbers for semi-N and semi-C tryptic peptides. Additionally, the FDR is the same for all samples due to the TMT sample mixing and multiplexing. We conclude that the search space expansion appears appropriate for the work presented here (which involves comparison between product- and substrate-based approaches), but it is possible that the model performance could be increased even further in future work by restricting the search space to only tryptic peptides.

- More detail is required regarding how data analysis is done. In the method section the procedure of fitting into linear model is explained from a theoretical point of view. It would be nice if the code that was used to generate the linear models could be added so that the method itself could be more easily taken up by the protease community. At least the linear model method should be explained in such a way that it can be reproduced. In addition, the generation of sequence logos is often done by readily available tools where peptide intensities are very often simply normalized to the abundance of the proteome. Since the data handling for the visualization of the sequence logos appear to be more sophisticated here, it would be advisable to provide an explanation of the algorithm used, or even better share the code.

We have followed the reviewer's suggestion and made the code available on GitHub (<https://github.com/fmi-basel/ggrossha-ProteaseSpecificity-qPISA>). We also discuss now in the Results (section: "Substrate versus product: a comparison"), the Discussion (section: "Experimental and analytical advances to identify protease substrate motifs"), and the Methods (section: "Product based linear modelling") the challenges of appropriate background selection for product (semi-tryptic) peptide analysis, why we consider abundance of the proteome insufficient, especially when analyzing interactions, and how substrate analysis simplifies this issue. (We refer to our answer to this reviewer's final question for additional detail.)

- Figure 1c: This is a scatter plot. Why the log₂ intensities on the axis are negative? Is this due to normalization? It is quite confusing to what one normally expects from a peptide abundance scatter plot. This panel could be made clearer.

The data plotted in this figure is median centered, explaining why there are negative values – we apologize for not having indicated this more clearly. In addition to our previous description in the Methods section (relevant part underlined: "The PeptideGroups table was imported into the peptide workflow in the in-house developed einprot R package version 0.7.0 (Soneson et al, 2023), [<https://github.com/fmicompbio/einprot>] to undergo log₂ transformation, sample normalization using the center.median approach, and imputation via the MinProb method."), we now also indicate the median-centering in the legend of Fig. 1C.

- There is no project named PXD042089 in the Pride database. This should be made

available for the reviewers and later for publication.

The project is currently private. We submitted access details to the journal along with the manuscript, but it seems that these were not transmitted to the reviewer – we apologize. The data can be accessed as follows: **Username:** reviewer_pxd042089@ebi.ac.uk
Password: 5gKCJ3A6. (Please log in using the left(!) of the two login-fields provided.) We will make these data public once the manuscript has been accepted for publication.

Minor points

- Easily addressable points
- Presentation and style
- Trivial mistakes

- The quality and layout of figure 1 could be improved

We assume the reviewer is referring to Fig. 1B; please see our answer concerning this point above.

- I somehow argue against the usage of the wording 'massively parallel' in the title or 'highly parallelized' in the main text. I believe this is a bit exaggerated, when the substrate specificity of only two proteases has been described in this manuscript.

We respectfully disagree with the reviewer since we are not referring to the number of proteases analyzed but to the number of peptides that we quantify and for which we find the attribute justified. Hence, we have opted to retain the title.

- About experimental design explanation, along the lines of my previous comment. In supplementary figure 1 the schematic provided does not help at all in the definition of biological (?) or technical (?) replicates. The figure legend text is also very poor and not helpful for the figure interpretation.

We have modified the text of the legend to better clarify what is shown, including indicating that the samples are technical replicates.

- This sentence: "this analysis would also compensate for certain compositional biases in the input material". What do you mean with that? Can you quickly elaborate in the text?

We agree that this is an important point, especially with the new data on semi-tryptic peptide analysis that we now include. Hence, we have added two new paragraphs in the Discussion to clarify this point:

“A major benefit of qPISA’s focus on substrates is that the tryptic peptides that do not change in intensity serve as an internal background in the modelling process, which can account for several experimental biases: First, the HeLa proteome has a biased amino acid composition, e.g. L has a prevalence of 9.9%, W of 1.2%. This issue is further exacerbated

when considering neighboring pairs of amino acids (as quantified by interaction terms), e.g., LL has a prevalence of 1.1%, WW only 0.02%. Secondly, the peptides in the HeLa tryptic digest are strongly depleted for K and R as a result of trypsin's specificity. Finally, peptides have varying detection rates in the mass spectrometer given their amino acid composition. Without a proper background, these biases would immediately skew quantitative assessment of enzyme specificity, in particular for the interaction terms. An appropriate choice of background is indeed a major challenge when analyzing semi-tryptic (product) peptides." (We further elaborate on this latter point in the newly added methods section describing how we performed modeling of product peptides.)

Reviewer #3:

Summary

The manuscript by Gudipati et al. explores the effectiveness of a method called Quantitative Protease Specificity Inference from Substrate Analysis (qPISA) in analyzing the activity of human dipeptidyl peptidase 4 (DPP4) and the *C. elegans* protease DPF-3, an orthologue of human DPP8/9. This methodology integrates the high-throughput Proteomic Identification of Cleavage Sites (PICS) with a combinatorial analysis of the interactions between different proteolytic subsite features that promote or inhibit cleavage. Using this approach, they developed a substrate prediction model for the studied proteases and demonstrated its ability to predict their substrate proteins' half-lives. Using this model, they examined DPP4's substrate specificity and demonstrated how to engineer one of its known substrates, GLP-1, a peptide with implications for diabetes and obesity treatment. They performed a similar analysis on *C. elegans* DPF-3, comparing its substrate specificity to that of human DPP4. This comparative analysis reveals both similarities and differences in substrate preferences between the two proteases. They also resolved the structure of DPF-3 using cryo-electron microscopy and compared it with the published structure of human DPP4, highlighting the structural features in the S2 and S1' subsites responsible for differences in cleavage preferences.

General remarks

This is a solid work that includes extensive proteomics, structural, and computational studies. The text is clear and easy to follow, although some experimental details lack clarity (see below). The key conclusions about the cleavage preferences and subsite cooperativity of DPP4—particularly the pronounced preference for P1=Ala—are important. These findings enhance our understanding of this enzyme, which may have future implications given the significance of this protease. However, this work does not represent a conceptual breakthrough, as it builds upon a previously published proteomics method (PICS) and reiterates the concept of subsite cooperativity previously presented (e.g., PMID: 31543345). It should be noted that PICS was The conclusions appear to be well-supported

by the data, aligning with and validating existing knowledge and hypotheses regarding DPP specificity, including some unpublished studies. The insights provided will undoubtedly be of interest to the community of protease biology researchers, particularly those focused on dipeptidyl peptidases.

We thank the reviewer for pointing out the relevance and quality of our study. We would like to emphasize that we have not claimed to be introducing subsite cooperativity as a novel concept, and the publication mentioned by the reviewer was cited in our introduction. Instead, we provide a tool to experimentally assess and quantify cooperativity. To clarify this further, we have now adapted the relevant section in the introduction as follows (new addition underlined): “Although this model implies that the interactions between the individual substrate residues and their corresponding subsites are relatively independent of one another, it is evident that constraints in size, geometry or charge may lead to interactions such that certain sequence combinations are favorable, others unfavorable, due to cooperative effects (Ng, Pike, and Boyd 2009; Qi et al. 2019) (Fig. 1A). However, quantifying such effects requires large data sets of quantitative cleavage information that have been difficult to obtain.”

Major points

*Novelty and concept of qPISA workflow

The simplified PICS protocol utilized in this study has been previously published several times (e.g., PMID: 27122596, PMID: 28315252, PMID: 28315250 with TMT, among others). The authors emphasize the advantage of using the quantification of PICS/qPISA input peptides compared to the quantification of cleaved peptides in PICS. This approach shifts the entire workflow to rely on the lack of signal (disappearance of the input peptides) rather than the appearance of signal (from the cleaved peptides), which provides direct evidence for the presence of cleaved substrates in the sample.

It is not clear how the authors utilize (if at all) the actual quantification of the different peptides in their workflow.

Can the authors demonstrate the advantages of their approach compared to the commonly used approach in PICS-like experiments, which defines specificity based on the identification and quantification of neo-terminal (non-tryptic) peptides? They should compare their results to those that will be obtained based on the non-tryptic peptides. Additionally, how effective is the suggested workflow, which relies on the disappearance of tryptic peptides, for determining the cleavage preferences of peptidases with broader specificity?

The qPISA workflow takes only the tryptic (input) peptides into account. However, we agree with the reviewer that it would be interesting to compare the results from our substrate analysis to those of a “product” analysis, and we have now included this (Fig. 3).

We point out that our approach involves two innovations: Analysis of substrate instead of product peptides, and quantitative modeling that, as the reviewer rightly notes above, foregoes the need for setting arbitrary cut-offs to define substrates. It is the combination

that makes it powerful and allows us to capture the interaction terms (as illustrated in Fig. 2B vs. 2C for the substrates). We show sequence logos for tryptic and semi-tryptic peptides in the Reviewer Figure 1 below, each of which does a good job in identifying suitable amino acids in P1 but neither of which can uncover protease subsite cooperativity, a key objective of our study. Hence, we went ahead and quantitatively modeled the product peptide changes including interaction terms, following the same approach as for the substrates. As detailed in the new section “Substrate versus product: a comparison” with the new Fig. 3, we find that this analysis yields qualitatively similar results to the tryptic peptide analysis but performs quantitatively less well. This is consistent with the fact that the fold changes of substrate peptides can be calculated more reliably because of a well-defined background (tryptic peptides that do not change in abundance). We conclude that qPISA outperforms product-based analysis.

We have not tested the approach on proteases with broader specificity, but would predict that the signal seen for the specific subsites and their interactions would be reduced. However, this is unrelated to this specific analysis (and would also be true if highly quantitative and large-scale analysis of products were achieved), and merely reflects the biology since broader specificity means that more diverse sequences can be accepted.

Reviewer Figure 1 (replotted from our response to Reviewer #2 above): Sequence logos using A) tryptic and B) semi-tryptic peptides. Panel A is identical to Fig. 2B in the manuscript. Panel B uses a total of 41,571 peptides, 3,600 product peptides as the foreground and 37,971 tryptic peptides as the background, selected as detailed in the Methods section under “Product based linear modelling”.

*Technical aspects in the used workflow

The peptide identification list from the DPP9 and DPF-3 experiment includes approximately 18% of peptides (~12,000 out of ~64,000) that were not quantified. This represents a significant portion of unquantified peptides, which is substantially higher than the typical 1-5% usually observed with TMT labeling. This discrepancy raises concerns about the efficiency of the labeling process or the optimization of the MS analysis settings and might

contribute to some biases in the results. The authors should perform a database search while setting all TMT modifications as variable, to demonstrate labeling efficiency and explain the reason for the high portion of missed quantifications.

It is unclear how the authors determined the intensity threshold for the different experiments and how they compared the log₂ intensity of the protease-treated samples to the no-enzyme control. Why did they not simply perform a statistical test to compare the ratios of protease to no-enzyme in order to identify significant changes? Additionally, the log₂ intensities of the tryptic peptides in the protease-treated samples, as reported in Suppl. Table 1, shows a broad distribution centered around zero. Can the authors explain the reasons for such a distribution?

We assume that the reviewer is referring to the number of peptides that we include in each analysis. We'd like to clarify how this number is determined:

all peptides in input table: 70398

remove peptides with ambiguous n terminus flanks -> 69701

remove peptides with low detection levels -> 56600

remove duplicate peptides -> 53499

These numbers show that there is no problem with missed quantifications, but that low detection levels led to exclusion of a number of peptides. The relevant log₂ intensity thresholds were set empirically, after inspecting the respective intensity distributions (Reviewer Figure 2, below), as is common in omics data. Note that these thresholds are set per experiment, not per sample or condition, ensuring equal treatment across conditions. We can also specifically confirm that TMT labeling and Cys alkylation efficiency were suitably high at >99.8% and >98.9 Cys, respectively.

Reviewer Figure 2: Peptide intensity distributions for (A) the experiment investigating both DPF-3 and DPP4 and (B) the DPF-3 replicate experiment. The red line indicates the cut-off used to remove peptides with low detection levels.

Concerning the identification of significantly changing peptides, we did not aim to distinguish changing peptides from non-changing peptides in a binary fashion using a statistical test. Instead, we calculated the \log_2 fold change between the buffer control samples and the enzyme-treated samples and devised linear models to predict that quantity. We added an additional description in the manuscript to clarify this point (new text underlined):

“To this end, we devised a linear model to predict the tryptic substrate peptide intensity changes in the experiment from their amino acid sequences. Linear models (described in more detail in Methods) use a set of independent variables as input to calculate an output such that the correlation between the output of the model and the observed data is maximized. This approach does not require the selection of potential substrates through a statistical test to distinguish changing peptides from non-changing peptides in a binary fashion. Instead, the model directly predicts the \log_2 fold change between the buffer control samples and the enzyme-treated samples. Its performance can be quantified by the R^2 , which is the amount of variability in the data that it can explain. In the following, to allow for a fairer comparison among models of different complexities, we use the adjusted R^2 , which accounts for different numbers of parameters in the models.“

Concerning the differential abundance analysis, the data in this table, plotted in Figure 1C, is median centered, explaining why they are distributed around 0 – we apologize for not having indicated this more clearly. We now updated our previous description in the Methods section to read (relevant part underlined): “The PeptideGroups table was imported into the peptide workflow in the in-house developed einprot R package version 0.7.0 (Soneson et al, 2023), [<https://github.com/fmicompbio/einprot>] to undergo \log_2 transformation, sample normalization using the center.median approach, and imputation via the MinProb method.”). We now also indicate the median-centering in the legend of Fig. 1C and in the table legend.

*Comparison of DPP4 and DPF-3

DPF-3 originates from *C. elegans*, and therefore, its target proteome composition and the resulting tryptic peptide library might differ from the human tryptic library used in this study. Can the authors discuss the potential impact of these differences, if any?

We agree that there will be differences in the proteome compositions, but it is indeed a key benefit of differential analyses, such as the one that we are using, that they compensate for compositional biases. In response to a request from reviewer #2, we now explain this point in more detail in the Discussion:

“A major benefit of qPISA’s focus on substrates is that the tryptic peptides that do not change in intensity serve as an internal background in the modelling process, which can account for several experimental biases: First, the HeLa proteome has a biased amino acid composition, e.g. L has a prevalence of 9.9%, W of 1.2%. This issue is further exacerbated when considering neighboring pairs of amino acids (as quantified by interaction terms),

e.g., LL has a prevalence of 1.1%, WW only 0.02%. Secondly, the peptides in the HeLa tryptic digest are strongly depleted for K and R as a result of trypsin's specificity. Finally, peptides have varying detection rates in the mass spectrometer given their amino acid composition. Without a proper background, these biases would immediately skew quantitative assessment of enzyme specificity, in particular for the interaction terms. An appropriate choice of background is indeed a major challenge when analyzing semi-tryptic (product) peptides.” (We further elaborate on this latter challenge in the Methods section describing the semi-tryptic peptide-based modeling that we newly performed according to the reviewers' requests.)

If a specific amino acid combination were absent from the human material, we would be blind to its effects, but as we discuss, we have a comprehensive coverage of di-peptides and the main limitation in our experiment stems from the reliance on tryptic lysates, which affects representation of K and R in P2 and P1. As also discussed, this can be rectified by use of a different lysate, e.g., Glu-C-treated. Finally, we added a clarification in the Discussion on when the use of lysates from various sources might be beneficial (new text underlined): “Different sources of biological material can also be used for lysate preparation, allowing use of lysates from the cell type, tissue or organism for which a user may wish to study the function of their protease of interest. Although not required to establish the general substrate specificity of a protease under investigation, this may facilitate identification of individual, biologically relevant substrates.”

*Other factors that might alter specificity

DPP4 and DPF-3 are relatively large proteins. Could other elements within their structures also contribute to their cleavage specificity shown in this work?

We agree that this is possible. Our preliminary investigation of the structures suggests that the two enzyme's substrate gating mechanisms may differ, which may explain why DPP4 appears to prefer small unstructured peptides as substrates whereas DPF-3 can also process larger proteins. However, we decided that the manuscript was quite complex already and without further experimental validation, the finding would remain somewhat speculative. Hence, in keeping within the scope of this study, we decided against including a more detailed speculation.

Minor points

* The proteomics data should be reported in detail in supplementary tables, including identified peptide sequences, scores, and quantification. It is recommended to follow the guidelines established in PMID: 23500130 or the HUPO guidelines available at: https://hupo.org/resources/Documents/HPPMSDataGuidelines_3.0.0.pdf

The raw proteomics data have been deposited, according to the guidelines, in the Pride repository (<https://www.ebi.ac.uk/pride/>) to facilitate distribution according to the FAIR principles. (Dataset identifier PXD042089, freely accessible upon publication of the manuscript, currently private but accessible using Username:

reviewer_pxd042089@ebi.ac.uk, Password: 5gKCJ3A6.) In addition to the raw data files, we also provide as part of this dataset two R data files (.rds) that contain meta data as well as the data that were created in the intermediate processing steps up to the point of obtaining the final normalized intensity values. The peptides selected in our analysis according to the criteria given in the Methods sections are available in Tables EV1 & EV4.

* Please add a supplementary table detailing the full sequences of the 21 peptides and their respective proteins used in Figure 3.

This comment appears to refer to Figure 3A (Fig. 2D in the revised manuscript), for which we have reanalyzed data from a previous publication, Keane et al. 2011 (doi:[10.1111/j.1742-4658.2011.08051.x](https://doi.org/10.1111/j.1742-4658.2011.08051.x)). We plot them below for the reviewer's convenience, but would consider it rather redundant to relist them in our publication.

Protein name	peptide sequence
GLP-1-amide	HAEGTFTSDVSSYLEGQAAKEFIAWLKGR
GLP-2	HADGSFSDMENTILDNLAARDFINWLIQTKITD
GIP	YAEGTFISDIAMDKIHQQDFVNWLLAQKGGKNDWKHNITQ
Glucagon	HSQGTFTSDYSKYLDSRRAQDFVQWLMNT
PHM	HADGVFTSDFSKLLGQLSAKKYLESLM
GRF-amide	YADAIFTNSYRKVLGQLSARKLLQDIMSR
Oxyntomodulin	HSQGTFTSDYSKYLDSRRAQDFVQWLMNTRNRNINIA
VIP	HSDAVFTDNYTRLRKQMAVKKYLNSILN
PACAP-amide	HSDGIFTDSYSRYRKQMAVKKYLAAVLGKRYKQRVKKNK
PYY	YPIKPEAPREDASPEELNRYASLRHYLNLVTRQRY
BNP	SPKMQVQSGCGFRKMDRISSSSGLGCKVLRHH
NPY	YPSKPDNPGEDAPAEDMARYSALRHYINLITRQRY
Substance P	RPKPQQFFGLM
CCL3/MIP1α	ASLAADTPTACCFYSYTRQIPQNFADYFETSSQCSKPGVIFLTRSRQVCADPSEEWQKYVSDLELSA
CCL5/RANTES	SPYSSDTPCCFAYIARPLPRAHIKEYFYTSKGCSNPAVVFVTRKNRQVCANPEKWWREYINSEMS
CCL11/eotaxin	GPASVPTCCFNLANRKIPLQRLESYRRITSGKCPQKAVIFKTKLAKDICADPKKWWQDSMKYLDQKSPTPKP
CCL22/MDC	GPYGANMEDSVCCRDYVRYRLPLRVVKHFYWTSDSCPRPGVLLTFRDKEICADPRVWVKMILNLSQ
CXCL2/Groβ	APLATELRCQCLQTLQGIHLKNIQSVKVKSPGPHCAQTEVIATLKNQKACLNPASPMVKKIIKMLKNGKSN
CXCL6/GCP2	VLTELCTCLRVTLRVNPKTIGLQVFPAGPQCSKVEVASLKNQKQVCLDPEAPFLKVIQKILDSGNKKN
CXCL9/MIG	TPVVRKGRCSICSTNQGTIHLQSLKDLKQFAPSPSCEKIEIATLKNQVQTCLNPDSADVKELIKWEKQVVSQKKKQKNGKHKQKVKLVKRSRQKKT
CXCL10/IP10	VPLSRTVRCISISNPVNPRSLKLEIIPASQFCPRVEIATMKKKGEKRCNLNPKSQAIIKLLKAVSKERSKRSP
CXCL11/ITAC	FPMFKRGRCLCIGPGVKAVKVDIEKASIMYPSNCDKIEVIITLKNKGQRCLNPKSQARLIKKVERKNF
CXCL12/SDF-1α	KPVLSYRCPCRFFESHVARANVKHLKILNTPNCALQIVARLKNNRQVCIDPKLKIWIQEYLEKALN

* In Figure 3 (B,C), please include the sequences of the (expected) product peptides. Were

other products identified? Please provide the tables for peptide identification and quantification as a supplement.

We have now included the sequences in the figure legend. (The Figure is now Figure 4 in the revised manuscript.) Concerning the second point, we note that we analyzed the peptides in pairs. One group contained GLP-1 wt (*HAE* (HAEGTFTSDVSR)) & GLP-1 H1D, E3P (*DAP* (DAPGTFTSDVSR)) and the other group GLP-1 H1D (*DAE* (DAEGTFTSDVSR)) & GLP-1 E3P (*HAP* (HAPGTFTSDVSR)). Since all peptides are identical between residues 4 and 12, we would have been unable to assign and further cleavages to specific substrates. Finally, we provide the requested data in the new Table EV6.

* What is COCharDiC (Page 2)? Could this be a typo for ChFradic? If so a different reference might be required (e.g. PMID: 26010716).

Thank you for catching our mistake, we meant indeed ChaFRADIC and have corrected the typo and included the correct reference.

12th Sep 2024

Manuscript Number: MSB-2024-12373R

Title: Massively parallel quantification of substrate turnover defines protease subsite cooperativity

Author: Rajani Gudipati

Dimos Gaidatzis

Jan Seebacher

Sandra Muehlhaeuser

Georg Kempf

Simone Cavadini

Daniel Hess

Charlotte Soneson

Helge Großhans

Thank you for sending us your revised manuscript. We have now heard back from the two reviewers who agreed to evaluate your study. As you will see below, Reviewer #2 is generally satisfied with the revisions but has raised a few minor issues. Reviewer #3, while recognizing the solidity of the work, noted that several key concerns, such as the broader applicability of the presented method beyond DPPs, remain unaddressed, which affects the overall impact of the study.

During our pre-decision cross-commenting process (in which the reviewers can make additional comments, including on each other's reports), Reviewer #2 agreed with Reviewer #3's concerns, stating "I share reviewer three's concerns regarding the novelty, the advancement over current methods, and the broader applicability of their approach. While the authors attempted to address these issues in their revisions, they failed to provide additional evidence to substantiate their claims. I also voiced my concern regarding the title, which I believe overstates the manuscript's contributions." The complete consultation notes are included below, following the Reviewer reports.

However, given the overall positive feedback from Reviewer #2 in their report and the acknowledgement from both reviewers of the work's solidity, we think we can give you an opportunity to address the remaining issues in a revision of your study. Specifically, please respond to Reviewer #3's criticisms regarding broad applicability of your method and we would strongly recommend providing some levels of evidence that demonstrates this broader applicability. At a minimum, any potential limitations in this regard should be discussed. Additionally, please ensure that the manuscript and title do not contain overstatements in light of the reviewers' comments. The other issues raised by reviewers need to be addressed as well.

On a more editorial level, please do the following:

1. Please remove the Authors Contribution section from the manuscript file.
2. Please move Funding information to the Acknowledgements section.
3. "Data and material availability" should be renamed to "Data availability". Please provide specific URLs for EMD-17582, 8PBA, PXD042089 datasets in the data availability statement. Please make sure the datasets are made publically available upon acceptance of the manuscript.
4. Callouts: missing callouts for Fig. 6B and Dataset EV7 (only mentioned in "Supplementary Material" section that needs to be removed from manuscript file); there is a callout for Suppl.Fig. 4, but no such figure - Please fix.
5. Appendix file needs to be provided in PDF format.
6. Please upload the Reagents and Tools Table as a separate .docx file. The template can be found in our author guidelines: <https://www.embopress.org/page/journal/17444292/authorguide#structuredmethods>.
7. Supplementary Material (and EV Dataset legends) should be removed from manuscript file.
8. Please provide the synopsis image in PNG format with the requested dimensions: 550px in width and a height between 400px and 600px.
9. Section order should be corrected: title page with complete author information, abstract, keywords, introduction, results, discussion, methods, data availability section, acknowledgements, disclosure and competing interests statement, references,

main figure legends, tables, expanded figure legends.

When you resubmit your manuscript, please download our CHECKLIST (<https://bit.ly/EMBOPressAuthorChecklist>) and include the completed form in your submission.

Please note that the Author Checklist will be published alongside the paper as part of the transparent process (<https://www.embopress.org/page/journal/17444292/authorguide#transparentprocess>).

If you feel you can satisfactorily deal with these points and those listed by the referees, you may wish to submit a revised version of your manuscript. Please attach a covering letter giving details of the way in which you have handled each of the points raised by the referees. A revised manuscript may be once again subject to review and you probably understand that we can give you no guarantee at this stage that the eventual outcome will be favorable.

I look forward to receiving your revised manuscript soon.

Kind regards,
Jingyi

Jingyi Hou, PhD
Scientific Editor
Molecular Systems Biology

We realize that it is difficult to revise to a specific deadline. In the interest of protecting the conceptual advance provided by the work, we recommend a revision within 3 months (11th Dec 2024). Please discuss the revision progress ahead of this time with the editor if you require more time to complete the revisions. Use the link below to submit your revision:

IMPORTANT: When you send your revision, we will require the following items:

1. the manuscript text in LaTeX, RTF or MS Word format
2. a letter with a detailed description of the changes made in response to the referees. Please specify clearly the exact places in the text (pages and paragraphs) where each change has been made in response to each specific comment given
3. three to four 'bullet points' highlighting the main findings of your study
4. a short 'blurb' text summarizing in two sentences the study (max. 250 characters)
5. a 'thumbnail image' (550px width and max 400px height, Illustrator, PowerPoint or jpeg format), which can be used as 'visual title' for the synopsis section of your paper.
6. Please include an author contributions statement after the Acknowledgements section (see <https://www.embopress.org/page/journal/17444292/authorguide>)
7. Please complete the CHECKLIST available at (<https://bit.ly/EMBOPressAuthorChecklist>).

Please note that the Author Checklist will be published alongside the paper as part of the transparent process (<https://www.embopress.org/page/journal/17444292/authorguide#transparentprocess>).

See also figure legend guidelines: <https://www.embopress.org/page/journal/17444292/authorguide#figureformat>

9. Please note that corresponding authors are required to supply an ORCID ID for their name upon submission of a revised manuscript (EMBO Press signed a joint statement to encourage ORCID adoption).

(<https://www.embopress.org/page/journal/17444292/authorguide#editorialprocess>)

Currently, our records indicate that the ORCID for your account is 0000-0002-8169-6905.

Link Not Available

11. Include a Reagents and Tools Table as part of the Methods section, which can be downloaded from our author guidelines (<https://www.embopress.org/page/journal/17444292/authorguide#structuredmethods>)

*** PLEASE NOTE *** As part of the EMBO Press transparent editorial process initiative (see our Editorial at <https://dx.doi.org/10.1038/msb.2010.72>), Molecular Systems Biology publishes online a Review Process File with each accepted manuscripts. This file will be published in conjunction with your paper and will include the anonymous referee reports, your point-by-point response and all pertinent correspondence relating to the manuscript. If you do NOT want this File to be published, please inform the editorial office at msb@embo.org within 14 days upon receipt of the present letter.

Reviewer #2:

The authors have made substantial efforts to address the concerns raised in the previous version of this manuscript, including those from Reviewer 3. They have provided a detailed and thoughtful response, particularly in terms of benchmarking their new method against traditional approaches. The authors have effectively demonstrated the advantages of their approach by using linear models to evaluate the variance in non-tryptic peptides. Their results, as shown in Figure 3, indicate that the substrate-based analysis performs better than the product-based analysis in explaining variance, which aligns well with the objectives of their study.

However, the original question regarding a classical, binary interpretation of the data-specifically using a log₂ fold change and a p-value threshold to generate sequence logos-was not directly addressed. While I understand that the number of peptide sequences included for sequence logo generation would differ based on the thresholds selected, and thus may not predict experimental data as effectively as the approach shown in Figure 3D, a comparison or discussion of this method would have been valuable.

Regarding the title, the authors clarified that their use of the term "massively parallel" refers to the number of peptides quantified rather than the number of proteases analyzed. While this clarification is appreciated, I believe the title could still be perceived as slightly exaggerated depending on the reader's perspective. However, this may ultimately be a matter of editorial discretion.

Overall, the revised manuscript is much improved, with the authors adequately addressing most of the concerns raised. Including a discussion on the classical binary approach would provide a more comprehensive comparison, but even as it stands, the manuscript is a strong contribution to the field.

Reviewer #3:

I appreciate the authors' efforts to address the revisions and provide additional information in response to my previous review. However, I still have significant concerns that remain unaddressed. While I continue to believe that the work presented in this manuscript is solid and deserves publication, I feel its impact and appeal will be limited to a small and specific audience, such as the DPP-researcher community.

My primary concern pertains to the novelty and usefulness of the suggested approach. The putative applicability of the qPISA approach is a key strength and one of the major "selling points" of this manuscript. However, without demonstrating its applicability for studies of proteases beyond DPPs, which are distinguished by their remarkable positional and sequence specificity, the impact of this manuscript will be significantly reduced.

I suspect that the unique specificity of DPPs may be the primary reason for the results shown in this work, and I would like to see experimental evidence demonstrating otherwise. For instance, the authors could apply their workflow to study caspase-3 or -7, which have a relatively well-defined cleavage sequence motif but lack positional specificity. This demonstration could be conducted with fewer repeats and without the extensive fractionation and MS runs used for the DPPs. Without such evidence, the authors' claims regarding the advantages of their approach and its adaptation for the characterization of other proteases are not fully supported.

Additionally, the "simplified" PICS procedure, which does not include enrichment, is not novel and has been used in several studies before (e.g., PMID: 27122596, which is cited in the text for a different purpose, PMID: 28315252, PMID: 28315250). This should be indicated more clearly in the text (Pages 6 and 17), where the lack of enrichment is presented as a novel aspect unique to the qPISA workflow.

The authors highlight two key novelties in their approach: the analysis of substrate peptides instead of product peptides, and the use of quantitative modeling that eliminates the need for setting arbitrary cut-offs to define substrates. While these features may indeed enhance the power of their method, I have concerns about the cost-effectiveness of the procedure compared to other methods for analyzing protease specificity. In practice, I believe that most users may opt for less extensive and expensive analyses, even if it requires using product peptide, arbitrary cut-offs, and less sophisticated modeling or statistics, as long as they can still obtain meaningful results (as the authors nicely show in the revised Fig.3).

Concluding Remark:

In light of these unresolved concerns, particularly regarding the limited applicability of the approach to proteases beyond DPPs and the questions surrounding cost-effectiveness, I regret to conclude that this manuscript does not meet the standards or scope of Molecular Systems Biology and should not be considered for publication in this journal.

Additional minor concerns that were not fully addressed:

While the authors have shared the raw data and a nice summary of MS analysis results, the PRIDE dataset lacks critical information necessary for re-analysis. Specifically, it is missing details on which sample corresponds to each of the TMT channels and the descriptions of the experiments represented by the RAW file names.

Reporting of the peptide identification tables is still lacking and includes only "processed" sequences. Please add a table/s with peptide sequences assigned, precursor charge and mass/charge, all modifications observed, score, protein accession number, and quantitative information (TMT-related information).

Pre-decision cross-commenting

Reviewer #3

I share some of the insights of Reviewer 2 regarding this manuscript. Reviewer 2 report addressed some important points and the modifications that the authors made following it are useful and improved the manuscript (as I also indicated in my report). However, I disagree with the comments concerning the "simplification of the PICS protocol by eliminating the need for enrichment". This simplified workflow was published several years ago by Oliver Schilling's lab (and cited in the same reference provided by Reviewer 2: <https://doi.org/10.1074/mcp.O115.056671>) and was also presented in, at least, two separate methods/protocols.

Moreover, as the authors acknowledged in their response to my initial review, the novelty of their approach lies in the analysis of substrate peptides rather than product peptides, along with their use of quantitative modeling, which eliminates the need for arbitrary cut-offs to define substrates. While I am aware of several protease-proteomic characterization studies that have also focused on substrates rather than cleavage products, I will credit the authors of this manuscript for being the first to base an entire workflow on this concept.

That said, my main concern with the presented approach is that it may not be broadly applicable to most proteases, and I believe Reviewer 2 did not sufficiently address this issue. In my view, these two central points-the applicability of the method and the novelty of the approach-are key factors that explain the differing overall ratings and recommendations between Reviewer 2 and myself.

I agree with Reviewer 2 that the authors did not adequately discuss the advantages and effectiveness of their "arbitrary cut-off"-free analysis compared to more established methods, such as log2 fold change and p-value thresholds. Taking these points, along with those outlined in my previous reports, into account, I, like Reviewer 2, acknowledge that this is a well-executed study. However, I find the novelty somewhat limited, and I believe its overall impact on the field of protease profiling and substrate identification will be small.

Reviewer #2

My revised evaluation of this manuscript focuses primarily on the biology of DPP4 substrate specificity and the structural analyses. The authors have now provided explanations for many details that were initially omitted regarding their analytical approach. However, I still share reviewer three's concerns regarding the novelty, the advancement over current methods, and the broader applicability of their approach. While the authors attempted to address these issues in their revisions, they failed to provide additional evidence to substantiate their claims. I also voiced my concern regarding the title, which I believe overstates the manuscript's contributions.

As for its suitability for publication, I believe this decision ultimately rests with the editor. The manuscript offers a solid characterization of the DPP4 protease biochemistry but does not (yet) demonstrate a truly novel method for substrate characterization.

Reviewer #2:

The authors have made substantial efforts to address the concerns raised in the previous version of this manuscript, including those from Reviewer 3. They have provided a detailed and thoughtful response, particularly in terms of benchmarking their new method against traditional approaches. The authors have effectively demonstrated the advantages of their approach by using linear models to evaluate the variance in non-tryptic peptides. Their results, as shown in Figure 3, indicate that the substrate-based analysis performs better than the product-based analysis in explaining variance, which aligns well with the objectives of their study.

However, the original question regarding a classical, binary interpretation of the data—specifically using a log₂ fold change and a p-value threshold to generate sequence logos—was not directly addressed. While I understand that the number of peptide sequences included for sequence logo generation would differ based on the thresholds selected, and thus may not predict experimental data as effectively as the approach shown in Figure 3D, a comparison or discussion of this method would have been valuable.

We thank the reviewer for their comments and apologize for having gotten ahead of ourselves concerning the sequence logos. We would like to explain: A traditional sequence logo, obtained by aggregating sequence information from product peptides identified by an arbitrary log₂-fold cut-off (i.e., a binary classifier) has no quantitative information on cleavage efficiency. (A peptide is either a substrate or not, but there is no distinction in the extent of its cleavage.) In other words, we cannot compare the performance of the approaches on this metric, because, by its very design, the binary classifier cannot predict activity quantitatively. Moreover, the traditional sequence logo considers only one position at a time. Hence, it cannot uncover interactions. However, these were the two key goals and achievements of our study: Developing an approach that allows us to assess cleavage quantitatively (and thus, for instance, obtain a tool to direct peptide engineering towards altered stability, as we show for GLP-1) and understand cooperativity among active pocket subsites. Hence, since we could not compare the two approaches on these metrics, we opted for what we considered the next best alternatives to the reviewer's request, i.e., sequence logos for product and substrate peptides based on the quantitative modeling but without interaction terms, which addresses the inability of simple sequence logos to show interaction terms, and full product-based modeling, to show that substrate-based quantification is superior to product-base modeling.

We have now amended Abstract, Results and Discussions in several places to clarify the distinctions between binary classifiers and the qPISA modeling approach, and the novel insights that qPISA enables. Additionally, and for the reviewer's reference, we are also including the requested figure below. As expected, a sequence logo (iceLogo) can reveal the major positive contributions of Ala or Pro in P1, but is difficult to interpret in the other positions. For instance, the ability of Pro in P1' to completely cancel out any cleavage stimulating activity of residues in P1 is not evident, nor is the specific interaction between Asp in P2 and Ala (but not Pro) in P1. We have decided against including the figure in the manuscript because we think that with our text edits, it is now obvious that these are shortcomings by design.

Reviewer Figure 1: Sequence logos (iceLogos) for A) tryptic peptides decreasing at least 2-fold and B) non-tryptic peptides increasing at least 2-fold upon incubation with DPP4. For reference, C) Fig. 2C from the manuscript is plotted representing the rich interaction terms revealed with qPISA.

Regarding the title, the authors clarified that their use of the term "massively parallel" refers to the number of peptides quantified rather than the number of proteases analyzed. While this clarification is appreciated, I believe the title could still be perceived as slightly exaggerated depending on the reader's perspective. However, this may ultimately be a matter of editorial discretion.

We appreciate that we have not been able to allay this concern of the reviewer and have therefore changed the title to "Deep quantification of substrate turnover defines protease subsite cooperativity"

Overall, the revised manuscript is much improved, with the authors adequately addressing most of the concerns raised. Including a discussion on the classical binary approach would provide a more comprehensive comparison, but even as it stands, the manuscript is a strong contribution to the field.

We again thank the reviewer for their strong endorsement and hope that we have now clarified any previously remaining open issues.

Reviewer #3:

I appreciate the authors' efforts to address the revisions and provide additional information in response to my previous review. However, I still have significant concerns that remain unaddressed. While I continue to believe that the work presented in this manuscript is solid and deserves publication, I feel its impact and appeal will be limited to a small and specific audience, such as the DPP-researcher community.

My primary concern pertains to the novelty and usefulness of the suggested approach. The putative applicability of the qPISA approach is a key strength and one of the major "selling points" of this manuscript. However, without demonstrating its applicability for studies of proteases beyond DPPs, which are distinguished by their remarkable positional and sequence specificity, the impact of this manuscript will be significantly reduced.

I suspect that the unique specificity of DPPs may be the primary reason for the results shown in this work, and I would like to see experimental evidence demonstrating otherwise. For instance, the authors could apply their workflow to study caspase-3 or -7, which have a relatively well-defined cleavage sequence motif but lack positional specificity. This demonstration could be conducted with fewer repeats and without the extensive fractionation and MS runs used for the DPPs. Without such evidence, the authors' claims regarding the advantages of their approach and its adaptation for the characterization of other proteases are not fully supported.

The reviewer defines protease specificity in two dimensions, sequence and position. "Positional specificity" in this terminology appears to refer to whether a protease has exopeptidase activity (such as the DPPs investigated by us) or endopeptidase activity (such as the caspases that the reviewer proposes to investigate). The reviewer reasons that it will be more challenging to identify motifs for endopeptidases using qPISA. We concur and had explicitly stated this point in every version of the manuscript reviewed by this reviewer, and devoted a full paragraph of discussion to this matter. Specifically, we explained that application of qPISA to endopeptidases would require identification of matching pairs of substrate and product peptides: the product would reveal the cleavage site, the substrate would provide the quantitative cleavage information. To explore feasibility more directly, we have now assessed how many such matching pairs we could identify in our dataset. Strikingly, among 3,600 semi-tryptic (product) peptides (those that we had also used for product-based modeling in Fig. 3), we could detect 2,348 matching substrates (65.2%). Hence, applying qPISA to endopeptidases seems feasible and beneficial with a sampling depth comparable to what we have shown here. We have now added a section describing these insights in the Results, pp. 11-12: "qPISA is particularly well suited for exopeptidases, where the cleavage position on a substrate is defined (e.g., two amino acids away from the N-terminus, in the case of DPP4). For endopeptidases, this information is not immediately available. However, it can be obtained from product peptides detected in the same experiment so that the cleavage site can be mapped onto the matching tryptic substrate peptide. As this approach would require a sufficiently large number of matched pairs of product and substrate peptides, we examined the extent of overlap in our assay. Notably, for the 3,600 tryptic peptides that we had identified as products, we found 2,348 matching substrates (65.2%), a number that is likely to increase with future improvements in the detection efficiency of mass spectrometers. Hence, we can obtain quantitative cleavage information from substrate peptide analysis while sacrificing only 34.8% of the detected cleavage events, indicating that qPISA can be beneficial also for the quantitative characterization of endopeptidases."

Additionally, the "simplified" PICS procedure, which does not include enrichment, is not novel and has been used in several studies before (e.g., PMID: 27122596, which is cited in the text for a different purpose, PMID: 28315252, PMID: 28315250). This should be indicated more clearly in the text (Pages 6 and 17), where the lack of enrichment is presented as a novel aspect unique to the qPISA workflow.

We thank the reviewer for this suggestions, we have now revised the relevant section to read “As pioneered by others (Biniossek *et al.*, 2016; Chen *et al.*, 2017; Tucher & Tholey, 2017), we omitted the enrichment for C-terminal or ‘prime-side’ cleavage products, and instead directly quantified peptides from treated and untreated control lysates using TMT-labeling and mass spectrometry.”

The authors highlight two key novelties in their approach: the analysis of substrate peptides instead of product peptides, and the use of quantitative modeling that eliminates the need for setting arbitrary cut-offs to define substrates. While these features may indeed enhance the power of their method, I have concerns about the cost-effectiveness of the procedure compared to other methods for analyzing protease specificity. In practice, I believe that most users may opt for less extensive and expensive analyses, even if it requires using product peptide, arbitrary cut-offs, and less sophisticated modeling or statistics, as long as they can still obtain meaningful results (as the authors nicely show in the revised Fig.3).

We believe that the reviewer misunderstood Fig. 3: the results involve the same type of quantitative modeling with interaction terms as the substrate analysis. A traditional binary classifier, by definition, ignores any quantitative cleavage information. Therefore, it does not pose a viable alternative when the objective, as in our case, is to predict cleavage efficiency quantitatively. We also point to our Discussion on the issues faced by product-based approaches concerning identification of a suitable background, which is particularly relevant when seeking to understand target site cooperativity. Similarly, a simple sequence logo cannot display interaction terms because it considers each position independently – for detail please refer to our response to ref. #2, and especially compare Reviewer Fig. 1B to 1C. With these improvements over the state of the art, we are confident that our approach provides opportunities to the field to ask new types of questions and that researchers in the field will appreciate the advance.

Concluding Remark:

In light of these unresolved concerns, particularly regarding the limited applicability of the approach to proteases beyond DPPs and the questions surrounding cost-effectiveness, I regret to conclude that this manuscript does not meet the standards or scope of Molecular Systems Biology and should not be considered for publication in this journal.

We are uncertain of how “cost-effectiveness” would be calculated, especially given that standard approaches cannot yield quantitative insights nor interaction terms, *vide supra*. At any rate, we refer the reviewer to the code that we made available on GitHub, which should facilitate adoption of our approach by the community.

Additional minor concerns that were not fully addressed:

While the authors have shared the raw data and a nice summary of MS analysis results, the PRIDE dataset lacks critical information necessary for re-analysis. Specifically, it is missing

details on which sample corresponds to each of the TMT channels and the descriptions of the experiments represented by the RAW file names.

The file "2003_2083_HeLa_DPPs_ExpDesign.txt" that has been part of the submission contained the desired information.

Reporting of the peptide identification tables is still lacking and includes only "processed" sequences. Please add a table/s with peptide sequences assigned, precursor charge and mass/charge, all modifications observed, score, protein accession number, and quantitative information (TMT-related information).

These data have all been available with the PRIDE deposition. Specifically, these files contain the full peptide tables with all the requested info, and sample annotations used in the einprot analysis:

L2_201119_JS_2003_HeLa_DPF3_TMT15plex_Fr_semiTrypsin_2MC_SPS65_NAc_PeptideGroups.txt

L2_210730_210804_JS_2083_TMT16plex_Fr_full_then_semiTrypsin_2MC_SPS65_NAc_PeptideGroups.txt

PD_TMT_L2_201119_JS_2003_HeLa_DPF3_TMT15plex_Fr_semiTrypsin_2MC_uniquePeptideGroups_Abundance_MinProb_center.median_limma_einprot0.7.0_sampleAnnot.txt

PD_TMT_L2_210730_210804_JS_2083_TMT16plex_Fr_semiTrypsin_2MC_uniquePeptideGroups_Abundance_MinProb_center.median_limma_einprot0.7.0_sampleAnnot.txt

The columns in question for the original peptide tables (_PeptideGroups.txt files) are:

Sequence: Sequence, Annotated Sequence (with flanking residues)

all modifications observed: Modifications

protein accession number: Master Protein Accessions

precursor charge: Charge by Search Engine Sequest HT

(theoretical precursor mass: Theo MHplus in Da)

mass/charge: m/z in Da by Search Engine Sequest HT

score: XCorr by Search Engine Sequest HT

quantitative information (TMT-related information): Abundances Grouped*

Pre-decision cross-commenting

Reviewer #3

I share some of the insights of Reviewer 2 regarding this manuscript. Reviewer 2 report addressed some important points and the modifications that the authors made following it are useful and improved the manuscript (as I also indicated in my report).

However, I disagree with the comments concerning the "simplification of the PICS protocol by eliminating the need for enrichment". This simplified workflow was published several years ago by Oliver Schilling's lab (and cited in the same reference provided by Reviewer

2: <https://doi.org/10.1074/mcp.O115.056671>) and was also presented in, at least, two separate methods/protocols.

Moreover, as the authors acknowledged in their response to my initial review, the novelty of their approach lies in the analysis of substrate peptides rather than product peptides, along with their use of quantitative modeling, which eliminates the need for arbitrary cut-offs to define substrates. While I am aware of several protease-proteomic characterization studies that have also focused on substrates rather than cleavage products, I will credit the authors of this manuscript for being the first to base an entire workflow on this concept.

That said, my main concern with the presented approach is that it may not be broadly applicable to most proteases, and I believe Reviewer 2 did not sufficiently address this issue. In my view, these two central points—the applicability of the method and the novelty of the approach—are key factors that explain the differing overall ratings and recommendations between Reviewer 2 and myself.

I agree with Reviewer 2 that the authors did not adequately discuss the advantages and effectiveness of their "arbitrary cut-off"-free analysis compared to more established methods, such as log₂ fold change and p-value thresholds. Taking these points, along with those outlined in my previous reports, into account, I, like Reviewer 2, acknowledge that this is a well-executed study. However, I find the novelty somewhat limited, and I believe its overall impact on the field of protease profiling and substrate identification will be small.

Reviewer #2

My revised evaluation of this manuscript focuses primarily on the biology of DPP4 substrate specificity and the structural analyses. The authors have now provided explanations for many details that were initially omitted regarding their analytical approach. However, I still share reviewer three's concerns regarding the novelty, the advancement over current methods, and the broader applicability of their approach. While the authors attempted to address these issues in their revisions, they failed to provide additional evidence to substantiate their claims. I also voiced my concern regarding the title, which I believe overstates the manuscript's contributions.

As for its suitability for publication, I believe this decision ultimately rests with the editor. The manuscript offers a solid characterization of the DPP4 protease biochemistry but does not (yet) demonstrate a truly novel method for substrate characterization.

18th Oct 2024

Manuscript number: MSB-2024-12373RR

Title: Deep quantification of substrate turnover defines protease subsite cooperativity

Thank you again for sending us your revised manuscript. We are now satisfied with the modifications made and I am pleased to inform you that your paper has been accepted for publication.

Kind regards,
Jingyi

Jingyi Hou, PhD
Scientific Editor
Molecular Systems Biology
